# Open-circuit and short-circuit loss management in wide-gap perovskite p-i-n solar cells

Pietro Caprioglio [1] ✉, Joel A. Smith [1], Robert D. J. Oliver [1,2], Akash Dasgupta [1], Saqlain Choudhary[1], Michael D. Farrar[1], Alexandra J. Ramadan [1,2], Yen-Hung Lin [1], M. Greyson Christoforo[1], James M. Ball[1], Jonas Diekmann[3], Jarla Thiesbrummel [1], Karl-Augustin Zaininger[1], Xinyi Shen[1], Michael B. Johnston [1], Dieter Neher [3], Martin Stolterfoht[3] & Henry J. Snaith [1] ✉

In this work, we couple theoretical and experimental approaches to understand and reduce the losses of wide bandgap Br-rich perovskite *pin* devices at open-circuit voltage ($V_{OC}$) and short-circuit current ($J_{SC}$) conditions. A mismatch between the internal quasi-Fermi level splitting (QFLS) and the external $V_{OC}$ is detrimental for these devices. We demonstrate that modifying the perovskite top-surface with guanidinium-Br and imidazolium-Br forms a low-dimensional perovskite phase at the *n*-interface, suppressing the QFLS-$V_{OC}$ mismatch, and boosting the $V_{OC}$. Concurrently, the use of an ionic interlayer or a self-assembled monolayer at the *p*-interface reduces the inferred field screening induced by mobile ions at $J_{SC}$, promoting charge extraction and raising the $J_{SC}$. The combination of the *n*- and *p*-type optimizations allows us to approach the thermodynamic potential of the perovskite absorber layer, resulting in 1 cm² devices with performance parameters of $V_{OC}$s up to 1.29 V, fill factors above 80% and $J_{SC}$s up to 17 mA/cm², in addition to a thermal stability $T_{80}$ lifetime of more than 3500 h at 85 °C.

The current most promising technological application of perovskite solar cells (PSCs) requires the integration of perovskite photovoltaic devices in a monolithic tandem architecture, either in Si-perovskite[1], all-perovskite[2,3] or CIGS-perovskite tandems[4]. Commonly, perovskite devices used in such tandem architectures are in the so-called "inverted" or positive-intrinsic-negative (*pin*) configuration, due to their superior stability and reduced optical losses when integrated into tandem devices. Additionally, these devices feature bandgaps wider (~1.7–1.8 eV) than typical perovskite cells optimised to maximise single junction efficiency (~1.5 to 1.6 eV). However, one major issue related to wide bandgap PSCs is the management of their energy losses. Firstly, compared to their narrower bandgap counterparts, wide bandgap

perovskites are known to suffer from significantly larger open-circuit voltage ($V_{OC}$) non-radiative losses, commonly limiting the $V_{OC}$ to the ~1.2 V range[3,5,6]. Historically, the origin of these $V_{OC}$ losses was proposed to be caused by halide-segregation and the consequent shift of the effective energy bandgap to lower values[7,8]. This phenomenon is connected to the high Br content required to achieve bandgaps above 1.6 eV and a resulting phase instability with these halide compositions[8]. However, more recently, this picture has been challenged by several studies which have demonstrated that halide segregation can only account for a small part of the overall $V_{OC}$ losses and that the limiting factor is mostly related to non-radiative recombination at the interface between the perovskite and the transport layers (TLs), before halide

[1]Department of Physics, University of Oxford, Clarendon Laboratory, Parks Road, Oxford, UK. [2]Department of Physics and Astronomy, The University of Sheffield, Hicks Building, Hounsfield Road, Sheffield S3 7RH, UK. [3]Institute of Physics and Astronomy, University of Potsdam, Karl-Liebknecht-Str. 24-25, D-14476 Potsdam-Golm, Germany. ✉e-mail: pietro.caprioglio@physics.ox.ac.uk; henry.snaith@physics.ox.ac.uk

segmentation has even commenced[9,10]. Accordingly, it has been shown that devices featuring methylammonium-free wide-gap perovskites, not exhibiting halide-segregation, still suffer from substantial $V_{OC}$ losses[11]. Importantly, the same study also showed that wide-gap PSCs display a significant mismatch between the internal quasi-Fermi level splitting (QFLS) achieved by the absorber and the external $V_{OC}$ of the device, the latter being substantially lower even for highly optimized and efficient devices. Additionally, strong $V_{OC}$ losses are commonly associated with a larger trap density in wide gap bromide-rich perovskite materials, limiting the potential of the absorber[6]. Another loss mechanism that wide-gap PSCs currently face is related to a reduction in short-circuit current ($J_{SC}$) under continuous illumination. A recent study demonstrated that ion redistribution in perovskite devices can induce screening of the internal field and limit the charge collection at short-circuit conditions, effectively limiting the power conversion efficiency (PCE) of the final device[12]. Therefore, maximizing the built-in voltage ($V_{bi}$) in the perovskite absorber has emerged as an imperative strategy to efficiently extract the photogenerated carriers at short-circuit conditions[13]. Accordingly, it has also been shown that perovskite devices exhibit substantial PL emission even at short-circuit conditions, meaning that charge extraction is not yet fully optimized and a significant fraction of charges still undergo recombination within the bulk of the perovskite absorber layer, even at short-circuit[14].

In this work, by coupling theoretical and experimental approaches, we investigate and effectively reduce these two major types of losses in *pin* wide-gap (1.8 eV) PSCs. At open-circuit conditions, we theoretically demonstrate that the $V_{OC}$ is mostly limited by a poor energetic alignment between the perovskite and the charge transport layers (CTL), causing a detrimental QFLS-$V_{OC}$ mismatch. Experimentally, we use guanidinium bromide (GuaBr) and imidazolium bromide (ImBr) to induce a low dimensional phase at the top surface, which eliminates the QFLS-$V_{OC}$ mismatch, improving the $V_{OC}$

of the device. At short-circuit conditions, we find that the $J_{SC}$ losses are related to poor charge extraction, which is consistent with internal field screening induced by mobile ions[15]. We find that modifying the hole transporting layer (HTL) with a series of ionic interlayers or by using molecularly thin self-assembled monolayers (SAMs) as HTLs enhance the charge extraction, which we infer to be due to maximizing the internal field at the perovskite/HTL interface, limiting the effect of the ionic screening. Our fully optimized devices, featuring a SAM as the HTL and ImBr as surface modifiers, allow for suppression of the QFLS-$V_{OC}$ mismatch, achieving $V_{OC}$ values as high as 1.29 V, and a strong reduction of the $J_{SC}$ losses, reaching $J_{SC}$s up to 17 mA/cm² and fill factors (FFs) above 80%, and corresponding efficiencies of 17%.

## Results and discussion
### Understanding the QFLS-$V_{OC}$ mismatch in wide-gap devices
In a recent study, we showed that wide-gap PSCs suffer from a significant mismatch between the internal QFLS and the external $V_{OC}$, even for highly optimized devices[11]. According to that study, as schematically represented in Fig. 1a, such effects can be caused by the energetic misalignment originating from using a wider perovskite bandgap with the same transport layers originally optimized for 1.6 eV devices. In Fig. 1b, drift-diffusion simulations of a series of perovskite *pin* devices with varying bandgaps show how the evolution of the $V_{OC}$ with respect to the perovskite bandgap is strongly affected by the energetic alignment of the transport layers. To understand the specific processes behind the open-circuit losses, we perform drift-diffusion simulations on a model 1.8 eV perovskite device utilizing phenyl-C61-butyric acid methyl ester (PCBM) as the electron transporting layer (ETL) and poly[bis(4-phenyl)(2,4,6-trimethylphenyl)amine] (PTAA) as the HTL, based on the results from Oliver et al.[11]. Simulation details can be found in Supplementary Note 1. For clarity, here we focus only on

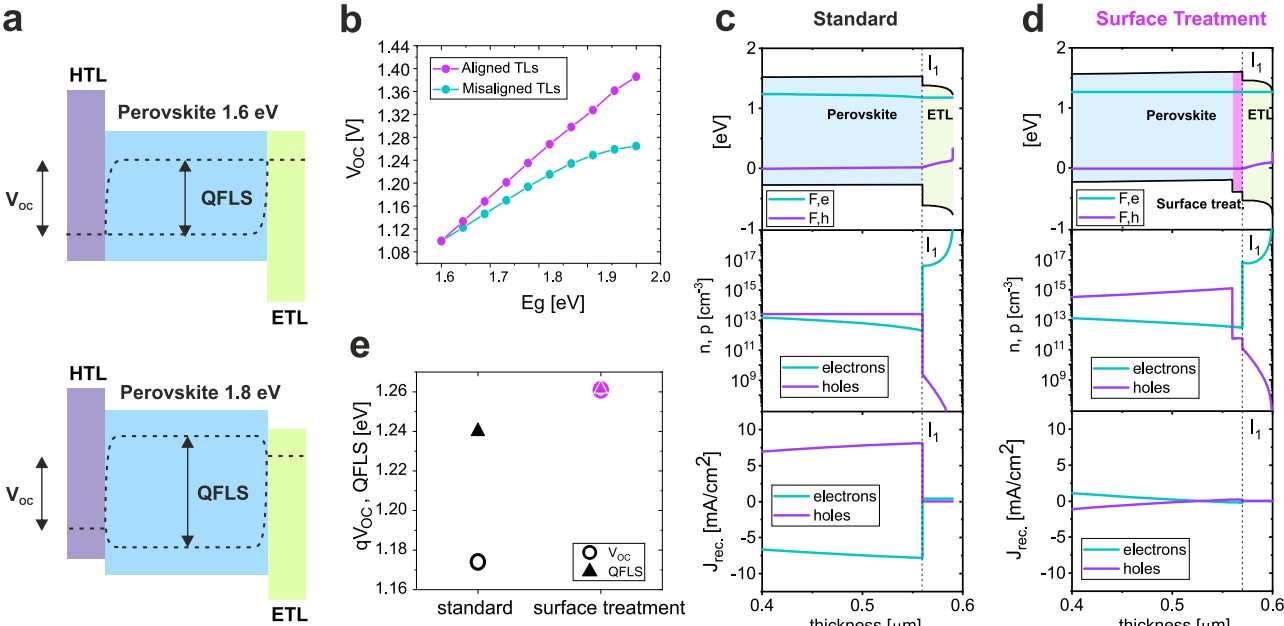

**Fig. 1 | Drift-diffusion simulations. a** Schematic diagram representing 1.6 eV and 1.8 eV perovskite solar cells using a hole and electron transport layer optimized for the 1.6 eV cell. The relation between the internal QFLS and external $V_{OC}$ is indicated by the dotted lines. **b** $V_{OC}$ obtained from drift-diffusion simulations for a perovskite solar cell device with varying energy gap. The purple data points represent a case where the energy levels of the transport layers are adjusted according to the perovskite bandgap in order maintain ideal alignment, whereas the turquoise points illustrate the effect of the energy levels remaining the same as those optimized for the 1.6 eV device. **c** Band structure and quasi-Fermi levels ($F_e$ and $F_h$) at $V_{OC}$ conditions close to the ETL side obtained from a drift-diffusion simulation of a standard 1.8 eV perovskite device. The respective hole and electron densities and the total hole and electron recombination currents ($J_{rec.}$) are also reported. **d** Band structure and quasi-Fermi levels at $V_{OC}$ conditions as in **c** for a 1.8 eV device implementing a modified perovskite where the surface at the ETL side features a lower valence band and wider bandgap. **e** QFLS and $V_{OC}$ relation from the devices simulated in **c**–**e**.

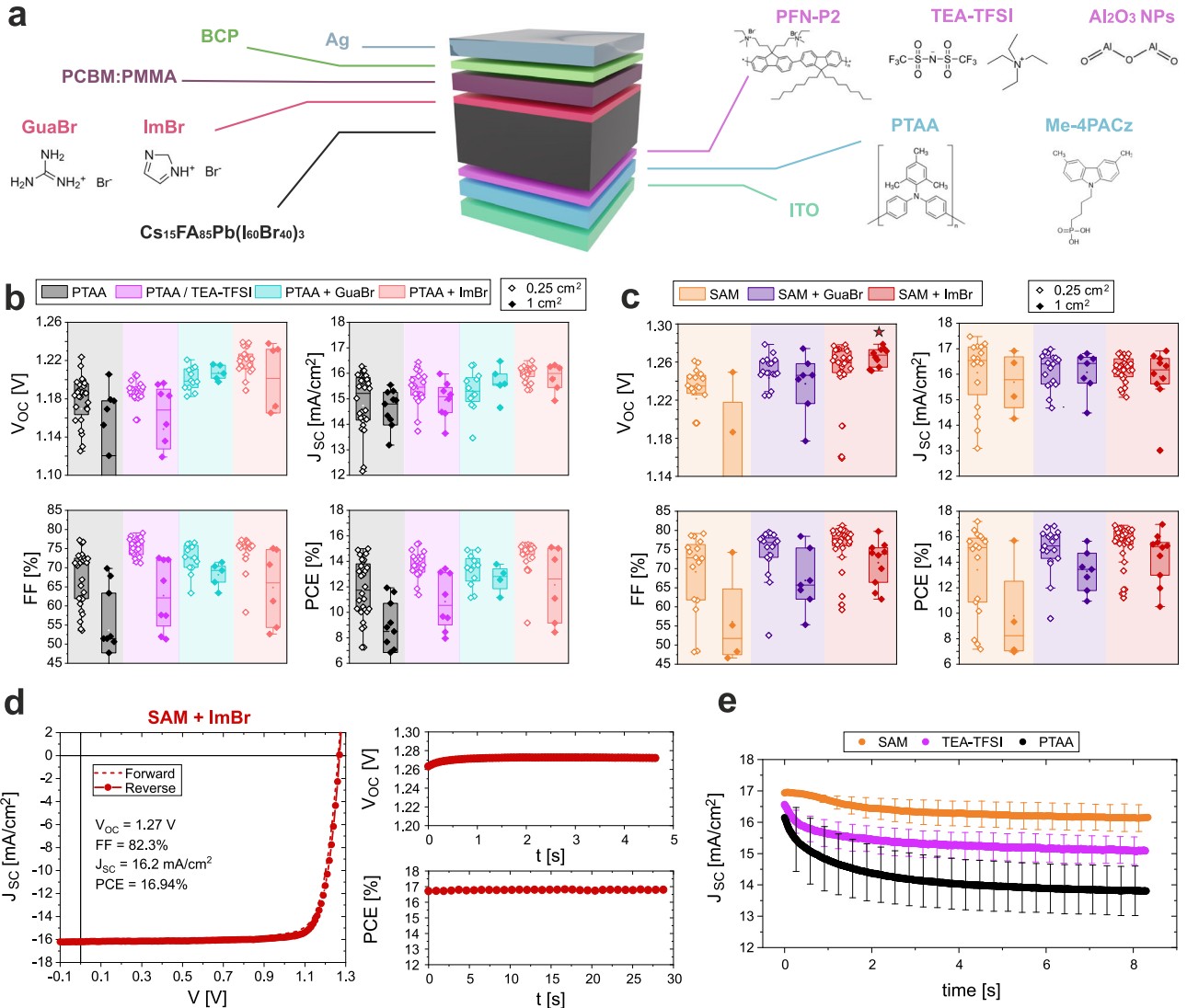

**Fig. 2 | Device characterization. a** Illustration of the typical 1.8 eV perovskite *pin* device used in this work. All the possible device architecture variations investigated in this study are also represented with their respective chemical structures. **b** Device statistic of a series of interface optimizations on perovskite devices using PTAA as the HTL obtained from reverse JV scans with a scan rate of 0.3 V/s. Here a PCE cut off was applied for devices with PCEs less than half of the mean value. **c** Performance metrics for surface-treated perovskite devices using Me-4PACz as the HTL, obtained from reverse JV scans with a scan rate of 0.3 V/s. Here a PCE cut off was applied for devices with PCE lower than half of the mean value. The star

indicates the record steady-state $V_{OC}$ for the 1 cm$^2$ device presented in Fig. S9. **d** Forward and reverse JV scans of the champion device using Me-4PACz as HTL, PCBM as ETL and ImBr as perovskite surface treatment. The JV parameters are reported for the reverse scans. The steady-state PCE under maximum power point (MPP) conditions is reported for 30 s. **e** Steady-state $J_{SC}$ decays over time averaged over 10–15 devices for each HTL. The measurement is taken by holding the device at $V_{OC}$ conditions and switching to $J_{SC}$ immediately afterwards. Error bars indicate the standard deviation.

discussing the perovskite/ETL interface, but the same argument can be translated to the HTL interface. In agreement with previous studies[16], we observe how, in a standard wide bandgap device, the combination of energetic misalignment between the perovskite and the ETL and fast interface recombination induces a significant mismatch between QFLS and $V_{OC}$, limiting the $V_{OC}$ to values below 1.2 V, as shown in Fig. 1e. In such a scenario, as shown in Fig. 1c, at open-circuit conditions, the energetic mismatch between the perovskite/ETL does not allow for the electrons reaching the PCBM layer to be reinjected into the perovskite. As a result, the large density of electrons in the PCBM can non-radiatively recombine across the interface with the free holes present in the perovskite at the proximity of the interface $I_I$, creating a constant depletion of carriers, causing the QFLS-$V_{OC}$ mismatch[16,17]. According to the Shockley-Read-Hall (SRH) formalism, the rate of trap assisted recombination ($R_{SRH}$) for unbalanced electron and hole densities, as is the case in the regions near this interface, will depend almost

exclusively on the minority carrier density, in this case, holes in the perovskite ($n_h$)[18]

$$R_{SRH} = C_h n_h \cdot N_I \propto n_h \qquad (1)$$

where $C_h$ is the hole capture coefficient and $N_I$ is the trap density. Details of the full derivation can be found in Supplementary Note 1. According to this picture, an efficient approach to reduce the non-radiative recombination at the perovskite/ETL interface consists of directly modifying the perovskite surface so that the hole density in the proximity of the PCBM interface is reduced, as presented in Fig. 1d. The whole thickness range of the simulation results is presented in Fig. S1. Consequently, by reducing the hole density at the interface $I_I$, recombination of charges across this interface is effectively nullified, despite the energetic alignment being in principle still detrimental. This is visible by the small recombination current at $I_I$ in Fig. 1d. As a

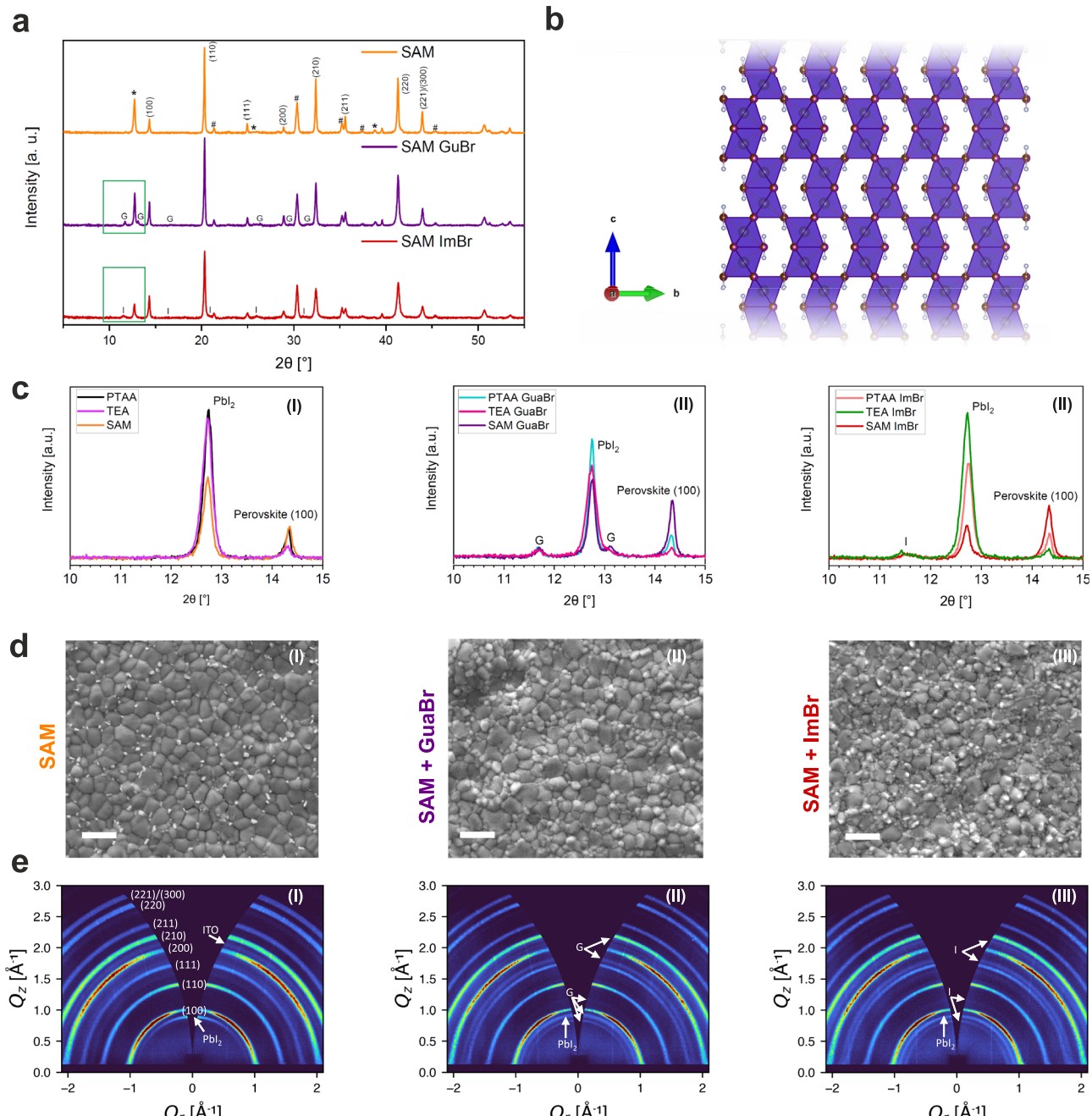

**Fig. 3 | Structural and morphological characterization. a** XRD patterns for neat SAM/perovskite films and after treatment with GuaBr or ImBr. Additional experimental details in **SI**. Indices are for the cubic perovskite phase, and secondary phases are marked: PbI$_2$ with *, ITO with #, GuaBr-induced phase with 'G' and ImBr-induced phase with 'I'. **b** Illustration of a mixed-halide 4H polytype, identified following the surface treatments (structure modified from reference Gratia et al.[33]). **c** Highlighted regions comparing XRD for all (i) neat, (ii) GuaBr-treated and (iii) ImBr-treated films, full XRD patterns for TEA-TFSI and PTAA samples is given in Figs. S17–18. **d** Scanning electron microscopy images of the perovskite surfaces in partially completed devices using (i) Me-4PACz and after treatment with (ii) GuaBr and (iii) ImBr (further SEM for all samples in Fig. S23), with scale bar = 1 μm. **e** 2D GIWAXS patterns for (i) SAM, **f** (ii) SAM GuaBr and (iii) SAM ImBr films, collected with a grazing incidence angle of 1°. 1D integrations of the data are given in Fig. S23.

result, as shown in Fig. 1e, the V$_{OC}$ is greatly enhanced, and the QFLS-V$_{OC}$ mismatch eliminated. Importantly, this approach allows us to access the internal thermodynamic potential of the wide-gap perovskite material without the need for optimising transport layers depending on the perovskite bandgap.

**Device optimization and J$_{SC}$ decays**

Based on the simulation results, we investigate a series of different optimization strategies for formamidinium (FA) caesium based perovskite, FA$_{0.85}$Cs$_{0.15}$Pb(I$_{0.6}$Br$_{0.4}$)$_3$, wide-gap *pin* devices, utilizing a PCBM:poly(methyl methacrylate) (PMMA) blend[19] as the ETL and PTAA treated with poly[(9,9-bis(3'-((N,N-dimethyl)-N-ethylammonium)-propyl)−2,7-fluorene)-alt-2,7-(9,9-dioctylfluorene)]dibromide (PFN-P2) as the HTL. Based on previous successful results, the large cation salts GuaBr and ImBr are investigated as surface modifiers for the perovskite top surface[20–23]. Notably, other Im-derivative salts have been successfully used as passivating agents for the perovskite bulk[24–28]. Optimization of both treatments can be found in Figs. S2–3. The JV

results, presented in Fig. 2b, show that for both surface treatments the $V_{OC}$ is significantly enhanced, reaching values up to 1.22 V for GuaBr and 1.24 V for ImBr. Concomitantly, the FF and $J_{SC}$ are also increased, suggesting better charge extraction from the top interface, resulting in devices with power conversion efficiencies (PCEs) above 15% in both cases. Interestingly, when comparing small area devices (0.25 cm²) to large area devices (1 cm²) the efficiency discrepancy originating from higher series resistance and potential inhomogeneity issues are greatly reduced for both surface treatments compared to the reference devices. Overall, the devices treated with the imidazolium salt slightly outperformed those treated with the guanidinium salt.

Next, we investigate the effects of a series of different ionic interlayers at the bottom interface between PTAA and perovskite, generally targeting cation-anion combinations which included various permutations of large immobile and small mobile ions. In Fig. S4, the PFN-P2 treatment applied on the PTAA surface in our reference devices is replaced with a bis(trifluoromethanesulfonyl)imide Li salt (Li-TFSI), or one of two quaternary ammonium compounds, tetra-methylammonium Br (TMA-Br) or tetraethylammonium TFSI (TEA-TFSI). Although similar salts have been used as passivating agents or to enhance device stabilities[29,30], in our study, we use a combination of anions and cations with different sizes to investigate the effects of mobile ($Li^+$, $Br^-$) or immobile ($TMA^+$, $TEA^+$, $TFSI^-$) ionic species at the interlayer between PTAA and perovskite. The device metrics presented in Fig. S4 reveal the superior interface enhancement when both ions are large/immobile, such as for TEA-TFSI. In Fig. 2b, devices using TEA-TFSI to treat the PTAA surface (optimized in Fig. S5), exhibit a significant enhancement in FF, exceeding 80%, and improvement in $J_{SC}$. Moreover, the TEA-treated samples exhibit a modest but beneficial increase in the $V_{OC}$ and overall better reproducibility, especially for large area devices. This might be attributed to a beneficial passivation effect of TFSI in contact with the perovskite, as recently demonstrated[22]. As shown in Fig. S6, the limited device performance when the TEA-TFSI is mixed into the perovskite bulk indicates that the improvement comes mostly from an interfacial effect rather than a modification of the perovskite absorber. To combine the beneficial effects of both optimized interfaces, GuaBr and ImBr are next applied in devices also utilizing TEA-TFSI. Surprisingly, as shown in Fig. S7, although the devices show consistently enhanced $J_{SC}$, the improvement in $V_{OC}$ originating from GuaBr and ImBr (in combination with TEA-TFSI), is not as beneficial as in the case of the standard PTAA devices, not improving the PCE further.

In another approach, we exchange the PTAA HTL with the SAM Me-4PACz[2]. Notably, in this case, to improve the wetting of the perovskite on the SAM, a highly diluted solution of $Al_2O_3$ nanoparticles is used as an interlayer[31]. Consistent with previous reports[2,32], the SAM devices reported in Fig. 2c significantly outperformed the PTAA devices, reaching $V_{OC}$s up to 1.24 V and $J_{SC}$s up to 17 mA/cm². In this new architecture, we again observe beneficial effects when GuaBr and ImBr are applied onto the perovskite surface. Similarly to the PTAA devices, the ImBr treatment is found to be consistently the most effective, greatly enhancing the $V_{OC}$ up to 1.28 V and the FF to above 81%. As shown in Fig. S8, for ImBr-optimized devices the use of the PCBM:PMMA blend is no longer necessary to increase the $V_{OC}$ and the removal of the PMMA exhibits devices with lower hysteresis. The resulting champion device shown in Fig. 2d exhibits a steady-state and JV-scan PCE of ~17%, a $V_{OC}$ of 1.27 V and a FF exceeding 82%. Notably, the fully optimized devices show an average PCE improvement of >3% absolute compared to the starting reference. Importantly, the ImBr treated devices again show a reduced performance gap between small and large area devices. Consequently, as exemplified in Fig. S9, also large area cells (1 cm²) are also able to reach a steady-state $V_{OC}$ of 1.29 V, a FF exceeding 80% and a JV-scan PCE of 17%. Example forward and reverse JV scans, and steady-state PCEs for all device configurations shown in Fig. 2b, c are presented in Fig. S10.

The effective device bandgap of 1.8 eV was estimated by using the inflection point of the external quantum efficiency $EQE_{PV}$ onset, Fig. S11. We did not observe a significant variation in bandgap energy regardless of the substrate on which the perovskite films crystallize (between 1.795 and 1.801 eV), as confirmed by absorption measurements in Fig. S12. Consequently, the $J_{SC}$ improvement observed in our devices, confirmed by $EQE_{PV}$ measurements in Fig. S13, is unlikely to originate from greater light absorption. Following a recent study by Thiesbrummel et al.[12], to confirm if the $J_{SC}$ differences are related to charge extraction issues, in Fig. 2e we investigate the time-dependent evolution of the current when switching the device from $V_{OC}$ to 0 V under light. We find that the magnitude of $J_{SC}$ decreases over time, suggesting a progressive reduction in charge collection efficiency. In agreement with the JV scans, modifying the PTAA surface with TEA-TFSI effectively reduces the $J_{SC}$ losses from >2 mA/cm² to about 1 mA/cm². Furthermore, using Me-4PACz additionally reduces the losses to only 0.5 mA/cm², predominantly maintaining the initial $J_{SC}$ over the measurement time. On the other hand, in the PTAA devices modifying the perovskite surface with ImBr reduces the degree of $J_{SC}$ decay to a small extent, Fig. S14, and shows no additional benefit in SAM devices, Fig. S15. Overall, this set of results suggests that the improved $J_{SC}$ originates mostly from enhanced charge extraction and that it can be tuned by modifying the transport layer interfaces, with a major effect on the HTL side. In order to check that this process is not related to any permanent degradation mechanism, we compared the $J_{SC}$ decays when measuring the same sample multiple times (Fig. S16). Consecutive measurements show identical behaviour, clearly demonstrating that the $J_{SC}$ decay is a fully reversible process and not related to any permanent effect. We will return to this point later in detail, to discuss the mechanism behind the reduction in $J_{SC}$ decay.

## Structural characterization

To elucidate structural changes induced by the GuaBr and ImBr treatments, we record X-ray diffraction (XRD) patterns for all films (SAM cases are shown in Fig. 3a, PTAA and TEA-TFSI are given in Figs. S17 and S18). The diffraction pattern for the untreated film confirms the primary perovskite phase, which we index as a cubic cell with lattice parameter 6.17(65) Å. We do not note any significant difference in the XRD patterns of the perovskite deposited on different HTLs. All films contain $PbI_2$ ($2\theta = 12.7°$) with comparable scattering intensity in the PTAA and TEA-TFSI films but reduced for the SAM films. Following the surface treatments, scattering features emerge at 11.7° and 13.1° in the case of GuaBr and at 11.5° for ImBr, related to new secondary phases. These peaks are consistent across all HTL cases, with the low-angle region for the different treatments highlighted in Fig. 3c. First, as shown in Fig. S19, we preclude the possibility of the treatment solvent inducing this phase behaviour with isopropanol (IPA) washing resulting in no secondary phase formation. Next, as shown in Fig. S20, we check if this data matched any previously identified guanidinium- and imidazolium-containing lead halide phase. However, none of these can account for the new peaks.

To identify these new phases, we increase the concentration of the GuaBr and ImBr treatment solution, resulting in stronger scattering intensity from both phases (Fig. S21 and Fig. S22). The new peaks following GuaBr treatment most likely correspond to a mixed halide 4H polytype phase (illustrated in Fig. 3b)[33], which is indexed as a hexagonal unit cell with lattice parameters $a = b = 8.77(57)$ Å, $c = 14.94(28)$ Å, $\beta = 120°$ and volume 996.6 Å³ (calculated XRD pattern is shown in Fig. S21). The ImBr treatment again leads to a polytype phase, which at higher concentrations can be indexed as a 4H polytype with lattice parameters $a = b = 8.84(73)$ Å, $c = 15.01(15)$ Å ($\beta = 120°$) with volume 1017.6 Å³ (shown in Fig. S22). Given that the peak positions of the 4H polytypes are consistent for different concentrations, the larger unit cell with ImBr treatment could indicate greater incorporation of the treatment cation. Deviations from the purely 4H character at the

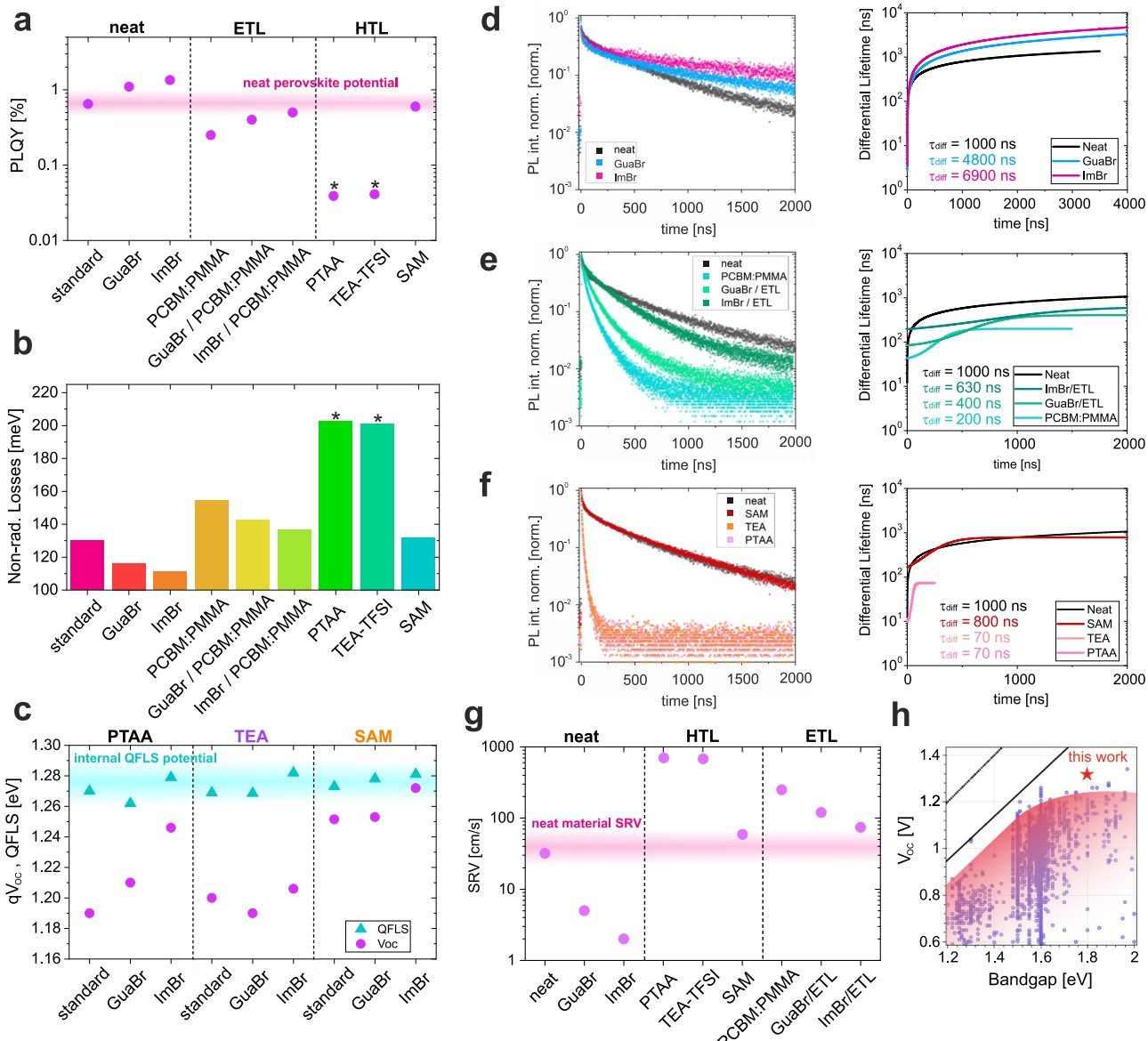

**Fig. 4 | Charge recombination and energy losses. a** PLQY results for different perovskite films and half-stacks (with either an ETL or HTL) illuminated with a 532 nm laser at a 1 sun equivalent intensity. The luminescence potential range of the neat perovskite on glass is highlighted in translucent pink. **b** Non-radiative recombination losses with respect to the radiative thermodynamic limit of the perovskite absorber calculated from the PLQY values reported in **a**. Details of the calculation can be found in **SI**. The asterisks in **a**, **b** indicate that the PLQY and non-radiative losses calculated for the PTAA and TEA-TFSI can be additionally affected by halide segregation, as discussed in Fig. S31. **c** QFLS-$V_{OC}$ comparison on full devices extracted from PLQY maps presented in Figs. S37–39. Each QFLS value is calculated from a PLQY measurement on the same pixel from which the $V_{OC}$ is

measured. The internal QFLS potential range of the devices is highlighted in translucent turquoise. **d** TRPL decays for different optimizations of perovskite films on (**d**) glass, (**e**) in contact with a PCBM:PMMA layer and (**f**) on different HTLS. The respective differential lifetimes for (**d**–**f**) are shown in the right panels. All TRPL decays presented in this figure are obtained by exciting with a 398 nm laser at a fluence of 15 nJ/cm². **g** Surface recombination velocities calculated from the differential lifetimes presented in **d**–**f**. Details of the calculation can be found in **SI**. **h** $V_{OC}$ vs. Bandgap literature comparison between our work and all perovskite *pin* cells reported in www.perovskitedatabase.com[36]. The black dotted line indicates the material bandgap and the black solid line the theoretical radiative $V_{OC}$.

extreme concentrations indicate a trend from face-sharing to corner-sharing octahedral connectivity at higher treatment concentrations in both cases. This observation is consistent with greater Br⁻ incorporation, with I⁻ generally preferring to occupy face-sharing sites in mixed-halide polytype phases[33].

These changes in the film composition with the surface treatments are commensurate with morphological differences we observe with scanning electron microscopy (SEM). Images of the three untreated film surfaces (SAM cases are shown in Fig. 3d, PTAA and TEA-TFSI in Fig. S23) show comparable morphologies with a densely packed perovskite grain structure in all cases, with the TEA-TFSI and

SAM samples exhibiting larger and more uniform grains. Additionally, brighter (electron-dense) platelet-like crystallites visible on the film surfaces indicates the presence of unreacted PbI₂. After treatment with GuaBr (Fig. 3d, **ii**), these features are no longer visible and instead small round grains have formed. In the case of the ImBr treatment, the surface structure has significantly evolved (Fig. 3d, **iii**), with the bright grains being replaced by small grains with different morphologies visible on the film surfaces. The effect of both treatments is consistent for each HTL (Fig. S23) and suggest the newly formed polytype material is primarily at the film surface. In Fig. S24, we also acquired cross-sectional SEM images of the films with each treatment, and as

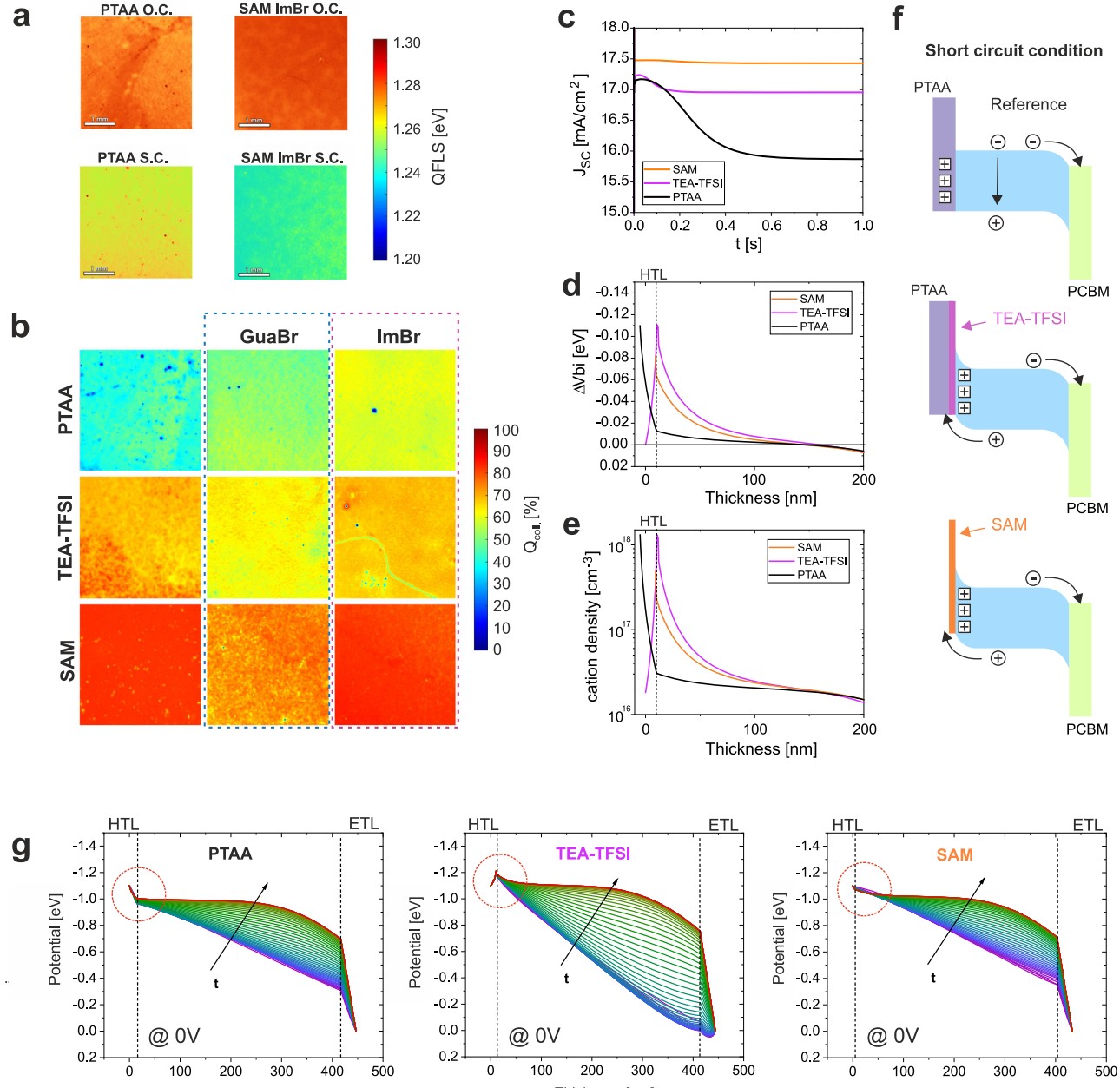

**Fig. 5 | Charge collection and field screening. a** QFLS mapping calculated from 3.7 × 3.7 mm PLQY images for PTAA and SAM-ImBr devices recorded at $V_{OC}$ and $J_{SC}$ conditions. **b** Series of 3.7 × 3.7 mm images obtained from PLQY images by illuminating a full device pixel under blue LED light at 1 sun equivalent. The images represent the quality of charge collection $Q_{coll}$ by comparing the PLQY at $V_{OC}$ and $J_{SC}$ conditions. **c** Time-dependent drift-diffusion simulation results presenting the decay of $J_{SC}$ over short timescales. **d** Variation of internal built-in voltage at 0 V from the perovskite bulk to the HTL at $t = 1$ s (quasi-steady state) for the devices shown in **c**. **e** The corresponding mobile cation density distribution for the same conditions presented in **d**. **f** Schematic representation of the energy bands for the simulated cases in **c**–**e** after the ion redistribution. **g** Simulated internal field profile distribution across the whole device thickness over the same timeframe presented in **c**. Red dotted circles highlight the field in the proximity of the HTL interface.

expected, we find that the perovskite bulk grain structure is unaffected by the surface treatments.

We investigate this surface effect further with grazing-incidence wide-angle X-ray scattering (GIWAXS) measurements, detailed in Figs. S25–26. The 2D diffraction pattern from a neat SAM film (Fig. 3ei) confirms the preferred <110> orientation of the 3D perovskite phase, as well as the typical out-of-plane orientation for the $PbI_2$. After surface treatment, we observe characteristic scattering from the GuaBr and ImBr-induced phases in agreement with the XRD measurements, with isotropic Debye-Scherrer rings showing these phases have no preferred orientation. Azimuthally integrated 1D plots of the GIWAXS data

(Fig. S26) confirm greater intensity from the secondary phases than in the XRD, consistent with a higher concentration of the secondary phase near the surface, as opposed to throughout the bulk of the film. Moreover, we also find that the remaining $PbI_2$ now has isotropic orientation, confirming that interaction with the post-treatments has resulted in a reorganisation of the film surface. In contrast, the perovskite orientation is unaltered by the surface treatment, indicating no significant evolution of the bulk structure in the films.

To confirm that the newly introduced phases are concentrated at the surface, we performed additional GIWAXS measurements at a range of incidence angles to probe the variation in polytype phase

intensity as a function of penetration depth, which are shown and described in Figs. S25–S27. These measurements clearly show both the ImBr and GuaBr-induced polytype phases are concentrated towards the top surface of the film.

X-ray photoelectron spectroscopy (XPS) measurements confirm that the surface composition of the GuaBr and ImBr-treated perovskite films is markedly different than that of the untreated films, as shown in Fig. S28. While guanidinium $(C(NH_2)_3)$ comprises similar chemical environments to formamidinium $(HC(NH_2)_2)$, compositional analysis from XPS (Fig. S29) confirms a greater proportion of N at the film surfaces after GuaBr treatment. For the ImBr-treated sample, the N 1$s$ and C 1$s$ high resolution spectra confirm additional nitrogen and carbon environments are present, as would be expected for the heterocyclic imidazole moiety (Fig. S28).

### Reduction of open-circuit losses

In order to study the non-radiative recombination behaviour in our devices, we investigate the photoluminescence quantum yield (PLQY) and time-resolved photoluminescence (TRPL) on a series of perovskite "half-stacks" and full devices. In Fig. 4a, the PLQY of the neat perovskite film of 0.7% is improved to 1% and 2% after GuaBr and ImBr treatments respectively, indicating effective surface passivation. Most importantly, the large PLQY losses induced by the PCBM layer are strongly reduced by either GuaBr or ImBr treatments, approaching similar PLQY values of the neat perovskite on glass. Consistent with the device $V_{OC}$s, ImBr outperformed GuaBr. Contrary to 1.6 eV devices[34], the PTAA/PFN-P2/perovskite interface shows a significant reduction of PLQY, denoting strong interface recombination at the HTL/perovskite interface. Consistent with the $V_{OC}$ of the full devices, the TEA-TFSI does not show any significant improvement. Importantly, as shown in Figs. S30–31, all half-stack perovskite films deposited on PTAA, present a significant degree of phase segregation and, therefore, the PLQY is estimated by fitting only the emission at 700 nm, as reported in a recent study[35]. As such, the losses and QFLS calculated from half-stacks on PTAA only represent an estimate of the non-radiative recombination losses. However, when the PTAA is replaced with the SAM, no halide segregation is observed during the measurement (Fig. S30) and the PLQY approaches that of neat perovskite on glass, denoting almost complete elimination of the interface recombination induced by the transport layers, consistent with previous findings for lower bandgap perovskites[2]. This phenomenon has been previously associated with more efficient charge extraction by the SAM layers compared to PTAA, which avoids a build-up of interfacial charge, responsible for phase segregation[2,36]. The associated non-radiative energy losses are presented in Fig. 4b, calculated with respect to the thermodynamic limit of the perovskite absorber as detailed in Figs. S32–S33. To check that the nanoparticulate $Al_2O_3$ "wetting layer" is not influencing the recombination at the SAM/perovskite interface, we investigate the PLQY and non-radiative losses for a SAM/perovskite sample with and without $Al_2O_3$ (Fig. S34). Given that the losses at the interfaces of the two samples are the same, we can conclude that the additional $Al_2O_3$ nanoparticle treatment is not influencing the recombination mechanisms of this interface and therefore not changing our conclusion about the role of Me4PACz as HTL. By measuring the PLQY of full devices it is possible to compare the internal QFLS and the external $V_{OC}$[16]. In Fig. 4c, we find that our reference device suffers from a very large QFLS-$V_{OC}$ mismatch of almost 100 meV. With regard to the reference device with PTAA/PFN-P2, treating the perovskite surface with GuaBr and ImBr helps to reduce the mismatch quite substantially. The devices with TEA-TFSI show only a modest decrease of QFLS-$V_{OC}$ mismatch and no effect upon GuaBr and ImBr treatment, in agreement with the device JV data in Fig. S7. Finally, when the HTL interface is also fully optimized by using the SAM, the combination with ImBr eliminates the QFLS-$V_{OC}$ mismatch entirely enabling to access the internal thermodynamic potential (for $V_{OC}$) of the perovskite absorber layer.

Now the $V_{OC}$ losses are comparable to the one of narrower bandgap perovskite, as highlighted in the literature comparison in Fig. 4h. Importantly, the experimental trend shown in Fig. 4c follows very consistently what we predicted from drift-diffusion simulations in Fig. 1f. Conceivably, the combination of larger Gua or Im cations with the Br halide at the perovskite surface leads to the formation of a lower dimensional phase with an enlarged bandgap at the surface, as proposed in several studies[37–39], resembling the simulation model in Fig. 1g.

The TRPL decays presented in Fig. 4d–f, consistently follow the PLQY trends in Fig. 4a. Intensity dependent TRPL measurements for each condition are reported in Fig. S35. The maximum differential lifetime[40], calculated in **SI**, of the neat perovskite of 1 μs is improved to 4.8 μs and 6.9 μs after GuaBr and ImBr treatment respectively. Importantly, GuaBr and ImBr surface treatment significantly increase the lifetime of the perovskite/ETL stack, confirming the effective reduction of interface recombination. The best case is found for the ImBr treatment, where the differential lifetime of the perovskite/ETL stack (630 ns) approaches that of the neat (un-passivated) perovskite film (1 μs). Similarly, the short lifetime in contact with the PTAA (70 ns) and the TEA-TFSI (70 ns) is dramatically enhanced using the SAM HTM (800 ns). We acknowledge that due to halide segregation in the PTAA and TEA-TFSI samples, these lifetimes might be influenced also by charge funnelling[35]. Finally, the SAM/perovskite stack exhibits almost the same lifetime as the neat perovskite (1 μs), consistent with the PLQY results. The quality of the Me4-PACz SAM/perovskite interface is consistent with recent findings which reveal minimal interfacial trap densities and negligible energy losses[2,36]. Accordingly, the calculated surface recombination velocities (SRV) are presented in Fig. 4g. Details of the calculation can be found in Supplementary Note 7.

### Reduction of short-circuit losses

In order to investigate the factors leading to the improved $J_{SC}$ of our devices and to investigate if there is any correlation with changes in the layer morphology, we perform PLQY and QFLS imaging of full solar cell devices under open-circuit and short-circuit conditions, a selection of which are presented in Fig. 5a (all series in Figs. S37–39)[41]. At open-circuit, no carriers are extracted from the device and hence the PL intensity is high. Under short-circuit conditions, we expect the PL to be significantly quenched due to the carriers being extracted from the solar cell and hence no longer being available for recombination within the active layer. Comparing these two operational conditions, we can observe that the different device optimizations have an impact on the remaining PL and hence QFLS at short-circuit conditions, indicative of significant free carrier density still present in the device during charge extraction. By comparing the PL at these two conditions we quantify the "quality" of charge collection of our devices. Following the approach detailed in Supplementary Note 8, in Fig. 5b a series of PLQY images represents the normalized collection quality $Q_{coll}$ for each device condition, expressed as

$$Q_{coll} = \frac{\text{PLQY(O.C.)} - \text{PLQY(S.C.)}}{\text{PLQY(O.C.)}} \qquad (2)$$

From these images, we observe that the device utilizing the standard PTAA/PFN-P2 HTL shows the lowest collection quality with $Q_{coll}$ of ~ 50%, consistent with the poor $J_{SC}$ in JV scans and the large $J_{SC}$ decay. We notice here that the bare PTAA sample also shows a significant amount of inhomogeneities over the probed area (3.7 mm$^2$) and spots with zero charge collection, possibly due to unconnected areas in the devices. Using GuaBr and ImBr improves this aspect, giving better homogeneity and slightly improved collection quality above 60%. The devices using TEA-TFSI instead of

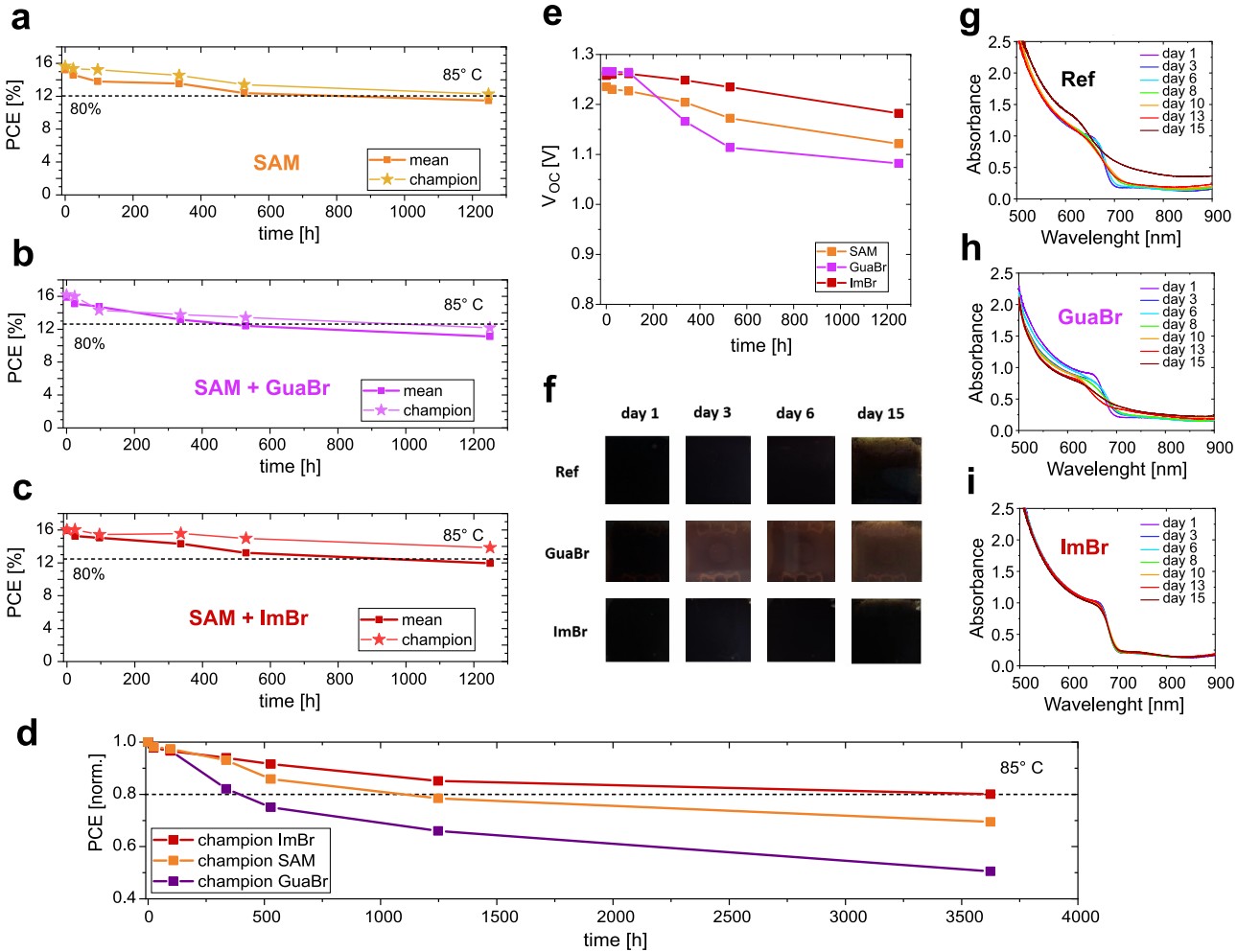

**Fig. 6 | Device and material stability. a–c** Thermal stability of unencapsulated devices using SAM as HTL with and without GuaBr and ImBr. The plots report the average PCE value of a series of devices for each condition. All devices were aged at 85 °C in N₂ atmosphere in dark conditions. **d** Long thermal stability at 85 °C in N₂ atmosphere in dark conditions of champion devices for each device condition.

**e** Average V$_{OC}$ values for the devices presented in **a–c**. **f** Photographs over a period of 15 days of the same samples presented in **d–f**. **g–i** Absorptance of unencapsulated perovskite films on glass with and without surface modification over a period of 15 days. The samples were aged in air at 20 °C with a relative humidity of 40–45% under ambient indoor light.

PFN-P2 exhibit overall a much more uniform morphology and increased charge collection to ~70% across all three cases. In line with the improved J$_{SC}$ results, the device implementing the SAM exhibits a better layer uniformity and $Q_{coll}$ of ~90%. These results agree with recent findings which revealed efficient hole transfer from the perovskite to the ITO by using the Me4-PACz SAM[36]. In the case of TEA-TFSI and SAM devices, the ImBr and GuaBr treatments do not increase the quality of charge collection, consistent with the JV results. Interestingly, similarly as for the J$_{SC}$ decays, we observe that the biggest impact on the charge collection efficiency originates from modifying the HTL interface rather than the perovskite surface. Overall, we find that the collection quality maps are qualitatively in very good agreement with the J$_{SC}$ trend obtained from the JV scans presented in Fig. 2, as well as with the J$_{SC}$ decays.

To establish a physical model of the enhanced collection efficiency, we perform steady-state and time-dependent drift-diffusion simulations implementing mobile ions on modelled PTAA, TEA-TFSI and SAM devices. The reference PTAA device has been modelled following a combination of experimental results and theoretical models previously published[11,13,17]. We modelled the TEA-TFSI device by adding a 1 nm interlayer with high ionic density ($10^{18}$ cm$^{-3}$) between the perovskite and the HTL, where the ions (positive and negative) were homogeneously distributed within this layer and fixed in position so

that they do not displace when an electric field is applied during simulated operation. For the SAM device, we simulated the SAM by using a 1 nm HTL, with the HOMO level set to align 0.1 eV shallower that the VBM of the perovskite and with lower interface recombination velocity (in comparison to PTAA/Perovskite). For all the models, we allow a certain density of ions from within the perovskite absorber layer to redistribute throughout the device layers under operation. We also investigated allowing both negative and positive ions to redistribute during measurements and found that in order to replicate the current loss at short-circuit that we observe experimentally, we need to allow only the positive ions to build up within the HTM layer. The details of the simulations can be found in Supplementary Note 10. In Fig. S40, the simulated JV curves are found to be in good agreement with the experimental trend. The time-dependent simulation results show that the simulated J$_{SC}$ decays, presented in Fig. 5c, qualitatively follow the experimental results presented in Fig. 2e with the different device configurations. We first examine the built-in voltage drop in the perovskite in the proximity to the HTL at $t = 1$ s and the variation of the mobile ion distribution (Fig. 5d, e). In our model, we assume that cations from the perovskite can diffuse into the PTAA HTL[42–44]. Consequently, we simulate that a large density (surface charge density of $10^{12}$ cm$^{-2}$) of cations accumulates within the PTAA layer, which consequently screens a large fraction of the available internal field in the

perovskite. On the other hand, despite that in the model all layers are permeable to ions, the high ionic density of the ionic-interlayer or the ultra-thin HTL (simulating the case of TEA-TFSI and SAM respectively), inhibits the diffusion of a large density of mobile ions into the HTL causing an accumulation of ionic charges at the perovskite/HTL interface and the formation of a narrow Debye layer[45]. As shown in Fig. 5d, these last two scenarios lead to a field at the $p$-interface in favour of hole extraction. We note that because of reduced interface recombination, the very thin HTL case exhibits the largest $J_{SC}$ even if the internal field in this region is slightly lower compared to the ionic-interlayer simulation. A schematic representation of the band diagram (simulated in Fig. S41) corresponding to these conditions is depicted in Fig. 5f. As presented in Fig. 5g, despite the difference in $J_{SC}$ decays between samples, we find that at 0 V, in all cases, the internal field in the bulk is screened over time due to ion redistribution. Despite that, in the case of the TEA-TFSI modification and the SAM, part of the initial internal field remains, promoting charge extraction. The correlation between the reduced $J_{SC}$ decays and the voltage drop at the $p$-interface indicates that the charge collection is strongly influenced by the extraction efficiency in the proximity of the HTL interface rather than in the perovskite bulk where the remaining field is small in all conditions. We also note that the upward band bending at the HTL interface, besides promoting hole extraction, also repels electrons, which can reduce non-radiative recombination of non-extracted carriers at short-circuit conditions.

### Impact of surface treatment on stability

Lastly, we investigate if there is any impact of the surface treatment and the lower dimensional phases induced by the GuaBr or ImBr on the stability of the perovskite films and devices. In Fig. 6a–c, we present the performance of devices over time, thermally stressed at 85 °C in $N_2$ in the dark. The GuaBr devices show a faster degradation than the "reference devices", with average PCE reaching 80% of its peak ($T_{80}$) after ~480 h, as compared to a $T_{80}$ lifetime of ~800 h of the SAM reference devices. The ImBr devices exhibited the highest thermal stability, reaching an average $T_{80}$ of ~950 h. This effect is clearly visible in Fig. 6e, where the $V_{OC}$ improvement of the ImBr device is maintained for >1200 h, whereas the $V_{OC}$ of the GuaBr devices falls below the reference cells after <200 h of thermal stress. We attribute these losses to an increase of non-radiative recombination after degradation of the perovskite absorber or interfaces. This GuaBr voltage loss is similar to what has been previously observed with phenylethyl ammonium passivation in $nip$ cells[46]. In Fig. 6d, we present the thermal stability of the champion devices for each different treatment scenario for longer times, and we find that, in the best case, the SAM-ImBr treated device retains 80% of its initial PCE for 3500 h.

As an alternative stress factor to temperature, water vapour and oxygen in the atmosphere can also lead to the degradation of perovskite films, and different treatments to the perovskite surface may have a positive or negative influence on this atmospheric stability. In Fig. 6f–i, we show photographs and UV-Vis absorption spectra of perovskite films with and without surface modification aged in air under ambient light and 40–45% relative humidity. We observe that the films which have been treated with GuaBr undergo a relatively rapid change in appearance, turning from dark brown/black to light brown, and a reduction in the strength of the absorption onset after only 3 days. In contrast, the ImBr treated films exhibit a predominantly unchanged absorption profile and characteristic black perovskite films after 13–15 days, whereares the reference films show a small change over this timescale. We show further images of aged perovskite films in Fig. S42. It, therefore, appears that the GuaBr treatment results in a deterioration of the ambient stability of the perovskite films, whereas the ImBr treatment results in a small improvement in the ambient stability of the perovskite films, in comparison to reference untreated films.

These results highlight that although in terms of device performance the two different surface treatments exhibit comparably beneficial effects, only the ImBr treated devices retained their initial improvement, enhancing the general stability of the perovskite material. Possibly, the $Gua^+$ cation can diffuse and incorporate in the bulk 3D perovskite structure more easily than $Im^+$, inducing additional degradation of the original perovskite phase. However, there could be a number of different causes, which are important to reveal in order to realise a generalisable strategy for passivation and long-term stability enhancement of metal halide perovskite materials and devices.

In conclusion, we have used a fundamental understanding of the operational device physics of wide bandgap PSCs to identify key losses in performance, and engineer improvement strategies to deliver enhanced widegap perovskite thin-films and solar cells. Specifically, we have rationalized that using either very thin HTM layers (SAMs) or introducing an ionic interlayer is important for reducing the short-circuit current of $pin$ perovskite solar cells. Introducing a surface passivation layer, which results in the growth of a low dimensional top layer, is important for reducing the voltage losses, eliminating the discrepancy between QFLS and $V_{OC}$, allowing access to the full potential of the absorber material. As such, we demonstrated that the major limitations of these devices are not related to the material itself but mostly to the transport layers and interfaces. Our results highlight how using optoelectronic modelling to guide device and interface engineering can lead to successful optimization of widegap PSCs, potentially leading to a more successful implementation in tandem solar cells with higher efficiencies.

## Methods
### Device fabrication

Glass substrates coated with patterned indium-doped tin oxide (ITO, Biotain Crystal Co., 10–15 ohm/sq. 1) were washed with acetone, Hellmanex III, deionised (DI) water and isopropanol (IPA). After UV-$O_3$ treatment (15 min for PTAA and 30 min for SAM), poly[bis(4-phenyl) (2,4,6-trimethylphenyl)amine] (PTAA) in a concentration of 1.5 mg/mL in Toluene was spin-coated at 6000 rpm for 30 s and immediately annealed for 10 min at 100 °C under $N_2$ atmosphere. Alternatively, a Me4-PACz (Tokyo Chemical Industry) solution 0.3 mg/mL in Ethanol was spin-coated at 2000 rpm for 30 s and immediately annealed for 10 min at 100 °C. The PTAA was additionally treated by dynamically spin coating a diluted solution (0.5 mg mL 1 in methanol) of poly[(9,9-bis(30-((N,Ndimethyl)- N-ethylammonium)-propyl)−2,7 fluorene)-alt-2,7-(9,9-dioctylfluorene)]dibromide (PFN-P2). Alternatively, the PTAA was treated with a tetraethylammonium- bis(tri-fluoromethanesulfonyl)imide (TEA-TFSI, Sigma-Aldrich) of 5 mg/mL in methanol using the same spin parameter of the PFN deposition. The SAM layer was additionally treated with a solution of $Al_2O_3$ nano-particles diluted at 1:150 by volume in IPA spin-coated using the same parameters of the PFN deposition. The perovskite layer was formed by spin coating a DMF:DMSO solution (4: 1 volume) starting at 1000 rpm for 5 s (ramping time of 5 s from stationary status) and then 5000 rpm (ramping time of 5 s from 1000 rpm) for 30 s. Before the end of the spinning process, a solvent-quenching method was used by dropping ethyl acetate (300 µL) onto the spinning substrates at 40 s after the start of the spin-coating process. The perovskite is annealed for a total of 50 min at 100 °C. The GuaBr and ImBr surface treatments were dynamically spin-coated on the perovskite after the first 30 min of annealing from an IPA solution of 2 mg/mL at 6000 rpm for 30 s. After deposition of GuaBr and ImBr the samples were annealed at 100 °C for 20 min (reaching a total of 50 min as the reference devices). A Phenyl-C61-butyric acid methyl ester (PCBM) solution (20 mg/mL in CB:DCB 9:1 by volume) was dynamically spun onto the perovskite layers at a speed of 2000 rpm for 20 s. The samples were then annealed at 100 ˚C for 3-5 min. After cooling down to room temperature, a bathocuproine

(BCP) solution (0.5 mg/mL in IPA) was dynamically spun onto the PCBM layer at a speed of 4000 rpm for 20 s, followed by a brief thermal annealing process at 100 ˚C for ~1 min. Both PCBM and BCP were processed inside the nitrogen-filled glovebox. The hybrid perovskite single-junction solar cells were completed by thermal evaporation of Ag electrodes (100 nm) through shadow masks under high vacuum ($6 \times 10^{-6}$ torr) using a thermal evaporator (Nano 36, Kurt J. Lesker) placed in ambient environment. The devices used for thermal stability uses Au electrodes instead of Ag.

### External quantum efficiency (EQE)
The external quantum efficiency of our devices was determined using Fourier transform photocurrent spectroscopy. Our custom-built set up is based on a Bruker Vertex 80 v Fourier transform interferometer. The solar cells were masked with a metal aperture such that the whole active area was illuminated by a tungsten halogen lamp[47]. To determine the EQE, the photocurrent spectrum of the device under test was divided by that of a calibrated Si reference cell (Newport) of a known EQE. The acquisition time for each photocurrent spectrum was ~60 s.

To determine the equivalent short-circuit current density under 1 sun irradiance from the EQE measurements, the overlap integral of the AM1.5 photon flux ($\varphi_{AM1.5}$) spectrum with the EQE was calculated. Explicitly, this is given by

$$J_{sc} = q \int_0 d\lambda \, EQE(\lambda)\varphi_{AM1.5}(\lambda)$$

where $q$ is the elementary charge and $\lambda$ is the wavelength.

### Time-resolved photoluminescence measurements
Time-Correlated Single Photon Counting was carried out using a 398 nm pulsed laser (PicoHarp LDH-D-C405M) with a repetition rate of 0.125 MHz for isolated films, and 2.5 MHz for those with transport layers attached. Photoluminescence was collected using the same monochromator, with a photon-counting detector (PDM series from MPD). Timing is controlled electronically using a PicoHarp300 event timer. PL decays were measured at the peak wavelengths of the PL spectra.

### Photoluminescence quantum yield
Excitation for the PL measurements was performed with a 532 nm CW laser (ThorLabs DJ532-10) through an optical fibre into an integrating sphere. The intensity of the laser was adjusted to a 1 sun equivalent intensity equal to 50 mW/cm², calculated for the excitation wavelength and the bandgap of the absorbing material. A second optical fibre was used from the output of the integrating sphere to an QEPro spectrometer. The system was calibrated by using a calibrated halogen lamp with specified spectral irradiance, which was shone into to integrating sphere. A spectral correction factor was established to match the spectral output of the detector to the calibrated spectral irradiance of the lamp.

### X-Ray diffraction characterization
1D-XRD patterns were collected with a Panalytical X'Pert Pro X-ray diffractometer in Bragg-Brentano geometry. The Cu source (K-α) was operated at 40 kV and 40 mA, with 0.04 radian Soller slits and typically 120 mins collection time per sample. Grazing-incidence wide-angle X-ray scattering (GIWAXS) data was acquired with a Rigaku SmartLab diffractometer. In this setup, a 3 kW Cu source (operating at 40 kV, 45 mA) with CBO-f optics was incident on thin films prepared as above. Samples were mounted on a 2D-XRD attachment head, which incorporated a knife-edge to prevent air scatter and an aligned beam stop for the direct beam. X-ray scattering was detected with a HyPix-3000 hybrid pixel-array 2D detector with a sample-to-detector distance of

65 mm. Sample widths were reduced to ~4 mm to minimise sample footprint broadening. Data was collected at a grazing incidence angle ($\alpha_i$) of 1° and the detector goniometer arm was rotated through 2θ angles from 0° to 40° in 1° steps. Detector images were resampled into Q-space and combined together using scripts based on the PyFAI and pygix libraries, which were also used for azimuthally integrated 1D profiles[48,49]. For variable-angle GIWAXS measurements ($\alpha_i = 0.2°-1.4°$), samples were measured using a different 2D-XRD aperture slit configuration with parallel-beam optics, a 1° in-plane parallel-slit collimator, a 0.1 mm incident slit, and 10 mm length-limiting slit at a single detector position, before processing as above. Powder XRD patterns and structural illustrations were calculated and rendered using VESTA[50].

### Scanning electron microscopy
An FEI Quanta 600 FEG scanning electron microscope was used to investigate perovskite layer morphologies with different surface treatments. The accelerating voltage was set to 5 kV and spot size was fixed at 2.5 (3.0 for cross-sectional images), with a typical working distance of 5 mm and 30 μs dwell time for single pass image acquisition. Focusing and alignment were done away from imaged areas to minimise electron beam–induced damage.

### X-Ray photoelectron spectroscopy
X-ray photoelectron spectroscopy measurements were carried out using a Thermo Scientific K-alpha spectrometer and a monochromated Al K-α X-ray source at a take-off angle of 90degrees. Core level spectra were recorded with a pass energy of 20 eV from an analysis area of 300 um × 300 um. The fitting procedure was carried out using CasaXPS, the background of the spectra were fitted using a Shirley lineshape and the peaks were fit using a mixture of Gaussian/Lorentzian (Lorentzian = 20%) line shapes.

### Current−voltage characteristic and $J_{SC}$ decay
Current density-voltage (J–V), maximum power point (MPP), and transient open-circuit voltage and short-circuit current density measurements were measured using a Keithley 2400 source measure unit (SMU) in ambient air under both light (simulated AM1.5 G irradiance generated by a Wavelabs SINUS-220 solar simulator) calibrated with a certified KG3-filtered reference diode (Fraunhofer) and dark. The active area of the solar cell was masked with a black-anodised metal aperture to either 0.25 or 1.00 cm². The forward J–V scans were measured from forward bias to short-circuit and the reverse scans were from short-circuit to forward bias, both at a scan rate of 0.3 Vs⁻¹. Active MPP tracking measurements using a gradient descent algorithm were performed to obtain the steady-state power conversion efficiency. The spectral mismatch factor was estimated to be 0.006 according to a previously reported method [https://doi.org/10.1039/C2EE03429H], and has been applied to calculate power conversion efficiencies. We estimate the systematic error of this setup to be on the order of ±5% (relative).

### Photoluminescence quantum yield mapping
The illumination was provided by a ThorLabs M450LP1 LED collimated by a Thorlabs SM2F lens. The emission was at 450 nm, well above the bandgap of the tested samples. The Intensity was controlled by controlling the power supplied to the LED. In order to determine the '1 sun' illumination, the sample was shorted and the LED power tweaked till the current readout was near the short circuit current measured on a solar simulator. The biasing was done using a Keithley 2400. More details are provided in Supporting Information.

### Thermal stability
Thermal stability measurements were carried inside an oven kept in the dark at 80 ˚C inside a glovebox under N₂ atmosphere.

**Drift-siffusion simulations**

Drift-diffusion simulations were carried out using the software SCAPS and SETFOS. Details on the simulations are given in the Supporting Information.

**Reporting summary**

Further information on research design is available in the Nature Portfolio Reporting Summary linked to this article.

## Data availability

The data that support the findings of this study are available from the corresponding author through the Oxford University open repository upon request.

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

## Acknowledgements

P.C. and H.J.S. acknowledge the ATIP (Application Targeted and Integrated Photovoltaics) Programme Grant funded by Engineering and Physical Science Research Council (EPSRC EP/T028513/1). R.D.J.O. and A.D. express their gratitude to the Penrose Scholarship for funding their studentship. A.R. acknowledge EU Horizon grant 861985 (peroCUBE). M.G.C. and J.A.S. acknowledge funding from the US Office of Naval Research (ONR) under award number N00014-20-1-2587. J.A.S. also acknowledges funding from EPSRC project EP/V027131/1. M.S. and D.N. acknowledge HyPerCells (a joint graduate school of the University of Potsdam and the Helmholtz-Zentrum Berlin) and the Deutsche Forschungsgemeinschaft (DFG, German Research Foundation)—project number 423749265 and 424709669—SPP 2196 (SURPRISE-2 and HIPSTER-PRO) for funding. M.S. further acknowledges the Heisenberg program from the Deutsche Forschungsgemeinschaft (DFG, German Research Foundation) for funding—project number 498155101.

## Author contributions

P.C. planned the experiments and interpreted the data. P.C. performed all the cell fabrication, photoluminescence and optoelectronic measurements, stability measurement, steady-state drift-diffusion simulations and wrote the manuscript; J.A.S. performed all the XRD and SEM analysis, assisted with manuscript preparation and contributed to the interpretation of the data; R.D.J.O. performed the TRPL measurements and analysis; A.D. performed PLQY imaging measurements and analysis; S.C. helped with the cell fabrication; M.F. helped with the photoluminescence measurements and cell fabrication; A.R. performed the XPS measurements and analysis; Y.H.L. provided insights for the cell fabrication; M.G.C. and J.M.B. helped during device testing and assembled the experimental setup; J.D. and J.T. performed the transient drift-diffusion simulations; K.A.Z. helped during device testing and provided assistance with experimental setups; X.S. helped with cell fabrication; M.B.J. supervised the TRPL measurements and results interpretation; D.N. and M.S. helped with the interpretation of the results and contributed in writing the manuscript; H.S. supervised the work, helped during the interpretation of the results and contributed in writing the manuscript.

## Competing interests

H.J.S. is the founder and Chief Scientific Officer of Oxford Photovoltaics, a company commercialising perovskite photovoltaics. The remaining authors declare no competing interests.
