## [Peer Review File · Nature Communications]

Open-circuit and short-circuit loss management in wide-gap perovskite p-i-n solar cellsREVIEWER COMMENTS

Reviewer #1 (Remarks to the Author):

In this work, Pietro et al reported a very comprehensive study on the energy losses of inverted perovskite solar cells, which is critical for obtaining high-performance tandem cells. Overall, this manuscript is well-written and the results presented here are interesting and potentially can be published in Nat. Commun. However, there are also many concerns that should be addressed carefully.

-In Figure S27, why does serious phase segregation happen in perovskite films deposited on PTAA substrate, not on SAM? Please explain this.

-What's the role of Al₂O₃ play in the SAM samples? Since the Al₂O₃ is an insulator layer, does it affect the conclusion of the SAM case? Please also do a comparison of SAM with and without the Al₂O₃ interlayer.

-The J_{sc} of all samples decreases very fast within a few seconds, like the 'burn-in' effect. Does this mean the devices fabricated by the authors' method are not stable? If not, please explain the device physics behind this phenomenon.

-In Fig S13, the EQE of the devices. Why does the PTAA device shows the red-shift in the band edge after 700nm wavelength? What's the physical meaning here? Usually, the blue part of the EQE spectrum reflects the optical information of the device. I saw that there is an obvious improvement around 400nm, please explain this.

-In Fig 3e, if the authors want to investigate the perovskite composition on the near-surface, GIWAXS should be conducted at very low incident angles. Please check this reference (Nat Commun 12, 3383 (2021)).

-Cross-section SEM of the perovskite films in Fig 3d should be compared as well.

-How does the 4H formed in the GuaBr and ImBr treated perovskite films? Does this caused by the solvent (e.g., IPA) effect? If so, the authors should also investigate the influence of different solvents on the surface composition of perovskite films after the treatment. For example, IPA can be replaced by chloroform (Energy Environ. Sci., 2019, 12, 2192--2199). The additional impurities should be avoided if the authors want to claim the improvements that arises from the modifiers. It is well known that the impurities, such as 4H or PbI₂ can bring detrimental to the carrier dynamics and stability in perovskite devices (<https://doi.org/10.1038/s41586-022-04872-1>).

-Can these materials for HTL treatments be applied to other metal oxides HTL, such as the widely used NiO_x (ACS Nano 2016, 10, 1, 1503–1511)? If so, it is also suggested to add one more condition to check the universality of this strategy.

Minors:

-Perovskite/CIGS tandem structure also shows promising applications (ACS Energy Lett. 2022, 7, 4, 1298–1307), which should be discussed on fair at the beginning of the Introduction.

-In fig 5f, please label each layer for different cases. It is difficult for all readers to understand it easily.

Reviewer #2 (Remarks to the Author):

This manuscript presents suggestive results for the development of wide-gap perovskite solar cells. The relationship between degradation and phase decomposition is shown. We can clearly see such degradation can be prevented by interfacial modification. Characterization of Photoluminescence Quantum Yield (PLQY) and Quasi-Fermi level splitting (QFLS) will help quantify energy loss at the interface and at the bulk in perovskite solar cells. Thus, the manuscript includes prominent results on both fundamental physics and practical applications, is worthy of publication. Please answer the following questions and correct any errors.

(1) What is your opinion of the effect of band edges (Urbach tail) on radiative recombination velocity and energy loss? Since the key issue of this paper is the effect of perovskite durability and degradation on its electronic states, I believe that this should be discussed in the manuscript in more detail.

When evaluating radiative recombination velocity, QFLS, and Voc losses in solar cells, the equations S4-S8 is usually used. As see in PL spectra shown in Figs. S27-S28 and absorption spectra suggested in Figs. 6g-i, in contrast, tailing states at around the wavelength of 700 nm were induced after degradation (decomposition of perovskite layers), which must causes variation in radiative recombination velocity since density of states near band edge severely influences the term of $EQE \cdot \Phi_{bb}$ in Eq. S5. Please check chap 6 in "Optical Processes in Semiconductors", J.I.Pankov, Dover. Although the authors discuss the relationship between QFLS and non-radiative recombination, I feel that without discussing optical properties, in particularly tailing states after degradation, the interpretation of the radiative recombination limit regarding solar cell properties will be unclear, since QFLS also changes with device state. In fact, PLQY is different for each device.

(2) Why did you use a repetition condition of 1 MHz of incident pulsed light for the TRPL measurement? If the light is irradiated before carrier recombination is complete, errors will occur in the carrier lifetime analytical results ($\tau \gg 1 \mu s$).

(3) error : Fig.5g (Description of the horizontal axis of the second graph) Thickness [eV] Thickness [μm]

(4) error : at lines 593, 523 nm CW laser  532 nm CW laser.

REVIEWER COMMENTS

Reviewer #1 (Remarks to the Author):

In this work, Pietro et al reported a very comprehensive study on the energy losses of inverted perovskite solar cells, which is critical for obtaining high-performance tandem cells. Overall, this manuscript is well-written and the results presented here are interesting and potentially can be published in Nat. Commun. However, there are also many concerns that should be addressed carefully.

We thank the reviewer for the overall positive feedback and for raising important points. We have provided a detailed response to all the comments below.

1. In Figure S27, why does serious phase segregation happen in perovskite films deposited on PTAA substrate, not on SAM? Please explain this.

Response: The authors thank the reviewer for the valuable discussion. The difference in phase segregation upon processing on PTAA vs. Me4PACz is indeed quite an important point, but this is not a new observation from our work. The explanation to this phenomenon has been clarified already by two earlier publications (<https://doi.org/10.1016/j.joule.2021.07.016>, DOI: 10.1126/science.abd4016). In those studies, the authors propose that there is significantly faster hole extraction when using a Me4PACz HTL as opposed to PTAA, avoiding a build-up of holes at the interface, which can drive phase segregation (<https://doi.org/10.1021/accountsmr.2c00076>, <https://doi.org/10.1002/adfm.202204825>). The faster hole extraction has been reaffirmed in the Joule publication by surface photovoltage measurements. In these past publications, the argument relating to hole build up has been verified by monitoring the PL shift characteristic of phase segregation for a set of Me4PACz/perovskite samples fabricated on glass or ITO. As predicted, the sample fabricated on glass, where no charge extraction from the HTL is possible, indeed showed evidence of phase segregation. On the other hand, the samples on ITO didn't show any phase segregation. Given that this mechanism has already been proposed in two publications, we did not comment extensively on this publication. However, we have highlighted this better with a more extended discussion in the manuscript.

"This phenomenon has been previously associated with the faster charge extraction by the SAM layers compared to PTAA, which avoids a build-up of interfacial charge, responsible for phase segregation."^{2,37}

2. What's the role of Al₂O₃ play in the SAM samples? Since the Al₂O₃ is an insulator layer, does it affect the conclusion of the SAM case? Please also do a comparison of SAM with and without the Al₂O₃ interlayer.

Response: The role of Al₂O₃ nanoparticles (NPs) is merely to increase the roughness of the substrate in order to enable better wettability for the perovskite precursor solution on Me4PACz. Note that the NP suspension is highly diluted 1:150 in IPA, so does not form a continuous layer, which as the reviewer points out would be insulating. As reported in this previous publication (DOI: 10.1126/science.abd4016) the chemical nature of the more hydrophobic surface of Me4PACz makes it difficult to process films on top from polar solvents such as DMF or DMSO. To confirm this point further, in a very recent publication, the authors use a substrate nanopatterning route to effectively avoid the same problem and note poor surface coverage and low device yield without using a rougher substrate (<https://www.nature.com/articles/s41565-022-01228-8>). However, we fully understand the concern of the reviewer and we investigated if the addition of an Al₂O₃ NP treatment on top of Me4PACz somehow changes the interfacial properties. Firstly, by processing the perovskite solution on top of Me4PACz with and without Al₂O₃ NPs we observe drastic changes in wetting (Figure R1 top). It is then clear that fabricating full devices on top of unmodified Me4PACz would significantly decrease the reproducibility

and reduce the yield of operational devices. Even though reproducibly fabricating devices on top of this layer is challenging, we managed to measure the PLQY of ITO/Me4PACz/perovskite samples with or without the Al₂O₃ layer by measuring the centre of the substrate. As indicated by the PLQY results, both samples show very similar photoluminescence efficiency, indicating very similar recombination losses (Figure R1 bottom). From the PLQY we can calculate the non-radiative losses associated with this interface (as described in the manuscript and SI) and we can therefore more accurately calculate the exact energy losses. Given that the losses at the interfaces of the two samples are the same, we can therefore conclude that the additional Al₂O₃ NP layer is not influencing the recombination mechanisms at this interface and therefore doesn't affect our conclusions about the role of Me4PACz as the HTL.

We included part of this discussion in the main text and added the figure to the SI.

“To check that the nanoparticulate Al₂O₃ “wetting layer” is not influencing the recombination at the SAM/perovskite interface, we investigate the PLQY and non-radiative losses for SAM/perovskite sample with and without Al₂O₃ (Fig. S35). Given that the losses at the interfaces of the two samples are the same, we can therefore conclude that the additional Al₂O₃ nanoparticle treatment is not influencing the recombination mechanisms of this interface and therefore not changing our conclusion about the role of Me4PACz as the HTL.”

Figure R1: a) Photographs of perovskite films processed on ITO/SAM/Al₂O₃ or ITO/SAM substrates. b) Corresponding PLQY and calculated non-radiative losses for the two films.

- The J_{sc} of all samples decreases very fast within a few seconds, like the ‘burn-in’ effect. Does this mean the devices fabricated by the authors’ method are not stable? If not, please explain the device physics behind this phenomenon.

Response: We thank the reviewer because this is again another very important aspect of the paper to be clarified. It is essential to understand that the J_{sc} decay process we are reporting and discussing throughout the paper is not related to any irreversible degradation mechanism. The J_{sc} decay does not imply that the devices are not stable at such short timescale, but they are representative of the effect of internal field screening due to ion motion at short circuit conditions. The penultimate part of our manuscript specifically proposes a model to explain such phenomena.

In order to clarify this point further, we compared the J_{sc} decays for the two extreme cases given in (PTAA and SAM), by measuring the same sample multiple times. From Figure R2, clearly, the J_{sc} effect is a reversible, “temporary” process and no permanent effect is observed. This is very much in line with an ion redistribution effect, which is a dynamic and reversible process that happens within seconds timescale. This type of measurement was inspired by this publication (<https://doi.org/10.1002/aenm.202101447>) where we observed and rationalised similar effects. We added further discussion of this in the main text and these new measurements in the SI.

“In order to check that this process is not related to any permanent degradation mechanism, we compared the J_{sc} decays when measuring the same sample multiple times (Fig. S16). Consecutive measurements show identical behaviour, clearly demonstrating that the J_{sc} decay is a fully reversible process and not related to any permanent effect.”

Figure R2: J_{sc} decay measurements of perovskite devices with PTAA or Me-4PACz SAM HTLs. Here “before” and “after” indicate multiple consecutive measurements to check the reversibility of the phenomenon.

4. In Fig S13, the EQE of the devices. Why does the PTAA device shows the red-shift in the band edge after 700nm wavelength? What’s the physical meaning here? Usually, the blue part of the EQE spectrum reflects the optical information of the device. I saw that there is an obvious improvement around 400nm, please explain this.

Response: We understand the reviewer’s concern and so we investigated this behaviour further. We originally attributed such small red-shifts of the band edge in PTAA devices to batch-to-batch variation and therefore did not comment further. Importantly, in Fig. S11 of the original submission, we estimated the bandgap from the 1st derivative of the EQE and all three samples show almost identical bandgaps of 1.8 eV. We calculated specifically for the EQE presented in Fig. S13 of the original submission and the bandgap is estimated to be 1.1798 eV for all samples, indicating that even the small red-shift observed in these specific EQEs does not change the absorber bandgap.

Figure R3: Bandgap estimation from $dEQE/dE$

However, to reconfirm the batch-to-batch variation argument we fabricated new TEA and PTAA devices following the same recipe and compared them to the SAM EQE (Figure R4). Indeed, we found this time that the previously observed red-shift is not present, as shown in the figure below. Therefore, we do not think that this has any significant implications for our study and does not require further investigation in this publication. We also note that the EQE data presented in Figure S13 is performed via Fourier transfer photocurrent spectroscopy. As can be seen, there is substantial noise in the blue region of the spectrum, therefore we do not want to overinterpret any small changes in this region. However, we have clarified these points in SI to avoid potential future confusion.

“We note a larger absorption at low energies for the PTAA device in this particular EQE series. We note that this small effect does not affect the absorber bandgap and is only due to batch-to-batch variation, as confirmed by the EQEs in Fig. S11.”

Figure R4: EQE curves for freshly prepared SAM, PTAA and TEA devices plotted on linear and logarithmic scales.

- In Fig 3e, if the authors want to investigate the perovskite composition on the near-surface, GIWAXS should be conducted at very low incident angles. Please check this reference (Nat Commun 12, 3383 (2021)).

Response: Due to limitations with our lab-based setup, to achieve the high-resolution 2D data shown in Figure 3 in the original manuscript, we employed a single incidence angle, corresponding to a penetration depth of 190 nm, where scattering will still be predominantly from the surface. To maximise

the intensity of the observed peaks and the resolution of the data in Q-space, data was acquired at a range of detector angles for around 6 hours of total measurement time.

Using a different configuration, we have performed additional GIWAXS measurements at a range of incidence angles from 0.2° to 1.4° to investigate the intensity of the polytype phase at estimated X-ray penetration depths (attenuation lengths) of 4 nm to 270 nm (as calculated in Figure S23 in the original submission). Example 2D scattering at the incidence angle of 0.2° for the ImBr-treated sample is shown below in Figure R5a. In a similar manner to the noted reference, we have azimuthally integrated the 2D data, subtracted the background and plotted the ratio of integrated peak intensity from the polytype phase peak at $2\theta \sim 11.5^\circ$ and the perovskite (100) peak at $2\theta \sim 14.3^\circ$ (Figure R5b). This data clearly shows that for both ImBr- and GuaBr-treated films, the polytype is concentrated towards the top surface as previously predicted. This discussion and new data have been added to the manuscript.

“To confirm the observations in the SEM that the newly introduced phases are concentrated at the surface, we performed additional GIWAXS measurements at a range of incidence angles to probe the variation in polytype phase intensity as a function of penetration depth, which are shown and described in Fig. S25-S27. These measurements clearly show both the ImBr and GuaBr-induced polytype phases are concentrated towards the top surface of the film.”

Figure R5: Variable incidence angle GIWAXS measurements of ImBr and GuaBr-treated ITO/SAM/perovskite samples measured at $\alpha_i = 0.2^\circ$ - 1.4° . a) Example 2D diffraction pattern for ImBr treatment measured at 0.2° incidence angle. b) The ratio of integrated peak intensity of polytype ($2\theta = 10.8$ - 12.4°) and perovskite phase ($2\theta = 13.5$ - 15.5°) peaks are plotted as a

function of calculated penetration depth (given in Figure S25) from 1D azimuthally integrated background-subtracted data. Errors in the penetration depth were calculated assuming an error on the incidence angle of $\pm 0.05^\circ$, and errors on the intensity ratio were calculated based on the standard deviation of the background noise in the 1D integrated data.

6. Cross-section SEM of the perovskite films in Fig 3d should be compared as well.

Response: We thank the reviewer for the valuable suggestion. We have acquired cross-sectional SEM on each of the three films shown in Fig 3d. From the cross sectional SEM images (Figure R6) we do not observe any significant changes to the grain structure or crystallinity of the perovskite layer. SEM is not the highest resolution imaging method, however, these images are consistent with the top surface treatments not significantly affecting the bulk region. We have now included these additional SEM measurements in the SI and additional text in the main manuscript.

“In Fig. S24, we also acquired cross-sectional SEM images of the films with each treatment, and as expected, we find that the perovskite bulk grain structure is unaffected by the surface treatments.”

Figure R6: Cross-sectional SEM images for ITO/SAM/perovskite samples with a) no treatment, b) GuaBr treatment and c) ImBr treatment, showing no alteration of the perovskite bulk grain structure with the treatments.

7. How does the 4H formed in the GuaBr and ImBr treated perovskite films? Does this caused by the solvent (e.g., IPA) effect? If so, the authors should also investigate the influence of different solvents on the surface composition of perovskite films after the treatment. For example, IPA can be replaced by chloroform (Energy Environ. Sci., 2019, 12, 2192--2199).

Response: In our original manuscript, we have already considered if this polytype formation could be an effect induced by the solvent treatment alone. As shown in Figure S18 of the original submission, we confirmed that this does not occur with solvent treatment by IPA alone. These salts are unfortunately insoluble in chloroform. We note the above reference highlights the unusual case of hexylammonium bromide solubility in chloroform, which is not the case for other alkylammonium bromides, or hexylammonium halides. The ImBr and GuaBr salts would be soluble in other alcohols such as n-butanol or ethanol, however IPA is perfectly representative of these other cases and due to only limited interaction with the perovskite layer components (<https://doi.org/10.1021/jacs.1c00757>) has been widely used for surface treatments. There may be small influences by choice of solvent, but this is likely to be of second order importance.

8. The additional impurities should be avoided if the authors want to claim the improvements that arises from the modifiers. It is well known that the impurities, such as 4H or PbI2 can bring detrimental to the carrier dynamics and stability in perovskite devices (<https://doi.org/10.1038/s41586-022-04872-1>).

Response: Concerning the impurity phases, the main results from our manuscript demonstrate that the presence of the 4H polytype on top of the 3D perovskite film is in fact very positive. The defect landscape and phase impurities in perovskites are complex and varied. In the above reference (using 17% Br, triple cation perovskites), the identified polytype is 2H, which as a 1D phase with completely face-sharing octahedral arrangement will be more susceptible to photodegradation by migration of organic cations. On the other hand, 4H polytypes are continuous 3D arrangements, which may inhibit

such material loss. We agree that in some instances, the interfaces between certain polytypes and 3D perovskites may induce increased defect densities, but this is not general and not consistent with our observations here. It is also important to notice that, in our study, we are forming a thin top surface layer of such phase and we do not alter the perovskite bulk.

Regarding PbI_2 impurities, the effect on carrier dynamics and stability is debated. PbI_2 defect clusters do not seem to participate in carrier-trapping (<https://doi.org/10.1039/D1EE02055B>). In FAPbI_3 films, PbI_2 forms latticed matched domains with the 3D perovskite phase, forming a low-defect interface which does not seem to compromise optoelectronic performance (<https://doi.org/10.1126/science.abb5940>). However, PbI_2 can itself photodegrade (Figure S34, <https://doi.org/10.1126/science.aba1628>) which does encourage us to reduce the excess PbI_2 stoichiometry employed here in future work.

On the stability of the heterostructure in this work, we found that ambient stability of the ImBr-treated films was superior to the control films, whereas GuaBr films suffered from worse ambient stability (Figure 6 in the main text). These results point to the cation composition of the polytype phase as being important in determining the overall film stability, rather than simply the formation of any polytype phase being beneficial.

9. Can these materials for HTL treatments be applied to other metal oxides HTL, such as the widely used NiO_x (ACS Nano 2016, 10, 1, 1503–1511)? If so, it is also suggested to add one more condition to check the universality of this strategy.

Response: We thank the reviewer for the valuable suggestion. Unfortunately, although our group has published high efficiency NiO_x containing cells in the past, we do not presently have a highly optimized NiO_x recipe capable of giving good devices with high reproducibility (this material appears to require a very dedicated researcher to optimise). However, we attempted to fabricate 1.8 eV devices using NiO_x as the HTL with and without TEA-TFSI treatment. Despite the generally low efficiencies, it is still possible to observe similar effects as for the PTAA devices (Figure R7). The J_{sc} and FF are slightly improved upon TEA treatment. Most importantly, by investigating the J_{sc} decays of these devices we observe the reduction of J_{sc} decays when the TEA layer is present. We can therefore conclude that, in principle, the TEA approach is also applicable to metal oxide HTLs. However, given the poor performance of NiO_x devices in our lab we do not think that including these data would improve our publication further. Exploring the TEA treatment on more robust and reproducible sputtered NiO_x could be a potential topic for a follow-up study.

Figure R3: Device performance metrics for ITO/ NiO_x (TEA)/perovskite/PCBM/BCP/Au devices. b) J_{sc} decay measurements averaging over all cells.

Minor: Perovskite/CIGS tandem structure also shows promising applications (ACS Energy Lett. 2022, 7, 4, 1298–1307), which should be discussed on fair at the beginning of the Introduction.

Response: Thank you to the reviewer for highlighting this omission. We have now included discussion of these results in the introduction.

Minor: In fig 5f, please label each layer for different cases. It is difficult for all readers to understand it easily.

Response: Figure 5f has now been modified to improve the clarity for the reader.

Reviewer #2 (Remarks to the Author):

This manuscript presents suggestive results for the development of wide-gap perovskite solar cells. The relationship between degradation and phase decomposition is shown. We can clearly see such degradation can be prevented by interfacial modification. Characterization of Photoluminescence Quantum Yield (PLQY) and Quasi-Fermi level splitting (QFLS) will help quantify energy loss at the interface and at the bulk in perovskite solar cells. Thus, the manuscript includes prominent results on both fundamental physics and practical applications, is worthy of publication. Please answer the following questions and correct any errors.

We thank the reviewer for the overall positive feedback on the manuscript. We have provided a point by point response to all their comments and suggestions below.

1. What is your opinion of the effect of band edges (Urbach tail) on radiative recombination velocity and energy loss? Since the key issue of this paper is the effect of perovskite durability and degradation on its electronic states, I believe that this should be discussed in the manuscript in more detail. When evaluating radiative recombination velocity, QFLS, and Voc losses in solar cells, the equations S4-S8 is usually used. As see in PL spectra shown in Figs. S27-S28 and absorption spectra suggested in Figs. 6g-i, in contrast, tailing states at around the wavelength of 700 nm were induced after degradation (decomposition of perovskite layers), which must causes variation in radiative recombination velocity since density of states near band edge severely influences the term of $E_{QE} \cdot \Phi_{bb}$ in Eq. S5. Please check chap 6 in "Optical Processes in Semiconductors", J.I.Pankov, Dover. Although the authors discuss the relationship between QFLS and non-radiative recombination, I feel that without discussing optical properties, in particularly tailing states after degradation, the interpretation of the radiative recombination limit regarding solar cell properties will be unclear, since QFLS also changes with device state. In fact, PLQY is different for each device.

Response: We thank the reviewer for the valuable discussion. Firstly, we would like to clarify that in our manuscript we do not investigate the effect of the perovskite degradation on the recombination processes. All the recombination analyses are done on freshly-made perovskite films. The study of the recombination mechanisms in our paper purely aims to relate the interfacial non-radiative losses to the actual performance of the fresh full device, specifically to their V_{OC} . It seems that the reviewer is asking how the change in absorption profile observed in the degraded films at the end our manuscript (Figs. 6g-i) is related to the decrease in V_{OC} observed while the full devices are aging under thermal stress. It is important to clarify here that the change in absorbance below the bandgap is due to scattering effects connected to the samples undergoing degradation in air and losing some material, resulting in a rougher film. Here changes in reflectance are not taken into account. From these measurements we cannot really conclude about changes in sub-bandgap absorption. We attribute the decrease in V_{OC} of the aged devices to an increase in non-radiative recombination processes, which have a far stronger influence on the V_{OC} compared to radiative recombination (related exclusively to optical properties of the material). To

confirm this point, we back calculated the PLQY for fresh and degraded devices, assuming a constant $J_{0,rad}$. The V_{oc} of the ImBr sample changes from 1.27 V to 1.25 V, with a respective change in PLQY from $1.7e-2$ % to $8e-3$ %. The V_{oc} of the GuaBr samples changes from 1.25 V to 1.09 V, with a respective change in PLQY from $8e-3$ % to $1.5e-5$ %. Finally, the V_{oc} of the Reference sample changes from 1.22 V to 1.12 V, with a respective change in PLQY from $2.5e-3$ % to $5e-5$ %. Such changes in radiative efficiency due to increased non-radiative recombination are perfectly feasible. On the other hand, if we take the most degraded sample (GuaBr) as an example and back calculate the corresponding theoretical change in $J_{0,rad}$ to have such an effect on the V_{oc} , we obtained a $J_{0,rad} = 7.5 e-18$ A/m² for the degraded device. This is 5 orders of magnitude higher compared to the $J_{0,rad}$ experimentally calculated for the fresh device of $1.55e-23$ A/m². Such a change in $J_{0,rad}$ is definitely not representative of our perovskite and it would require a change in absorber bandgap of several eVs and is impossible to achieve from sub-bandgap states only. Moreover, given that we still see the V_{oc} changes for the ImBr sample - which doesn't show any changes in absorption profile after ageing - we can conclude that the increase V_{oc} losses after degradation are due to increase in non-radiative processes. We added:

“We attribute these losses to an increase of non-radiative recombination after degradation of the perovskite absorber or interfaces.”

We also calculated the $J_{0,rad}$ for each of our different device architecture freshly prepared before aging. By looking at the tail state absorption from our different devices we observe that they all have almost identical contributions in absorption (Figure R8 right). Consequently, they all present very similar $J_{0,rad}$ values and, respectively, $QFLS_{rad, PTAA} = 1.496$ eV, $QFLS_{rad, TEA} = 1.499$ eV and $QFLS_{rad, SAM} = 1.499$ eV. Calculating individual radiative QFLS for each device shows that the variation is only on the order of 3 meV and therefore negligible. Therefore, we can conclude that the influence of the tail absorption to $J_{0,rad}$ and consequently on QFLS is almost identical for each device type investigated and do not influence the recombination analysis in this study. In the original manuscript, previously we omitted this discussion to not create redundancy. We have now implemented this discussion and this additional plot in the SI.

“Comparing the absorption of our different devices we observe that they have almost identical contribution from tail states. As such, all devices result in very similar $J_{0,rad}$ values and the following $QFLS_{rad, PTAA} = 1.496$ eV, $QFLS_{rad, TEA} = 1.499$ eV and $QFLS_{rad, SAM} = 1.495$ eV. Calculating individual radiative QFLS for each device shows that the variation is only in the order of 3 meV and thus negligible. Therefore, we can conclude that the influence of the tail state absorption to $J_{0,rad}$ and consequently on QFLS is almost identical for each device type investigated in our study.”

Figure R 4: General $J_{0,rad}$ calculations method from EQE and BB radiation for an exemplary 1.8 eV device (left). $J_{0,rad}$ calculations for each of our 1.8 eV perovskite using different device architectures.

Regarding the PL spectra showing phase segregation (Figs. S27-S28), (half-stack films on PTAA) we can indeed only provide an estimate of the PLQY and QFLS, as already indicated in the original manuscript. This is related to the fact that now part of the emission also comes from segregated domains, and we apply a peak fit and deconvolution in order to isolate the emission from the widegap material. This could already introduce some level of uncertainty. Moreover, if this new phase also significantly contributes to the absorption, we would indeed have a different $J_{0,rad}$ influencing the QFLS and the situation would become more complicated. Note that in one of our previous works, we investigated thoroughly exactly this aspect, specifically the relation between phase segregation and the reciprocity relation, as well as its impact on QFLS calculations (<https://doi.org/10.1021/acsenergylett.0c02270>). We have clarified this point better in the manuscript to avoid further confusion.

“As such the losses and QFLS calculated from half-stacks on PTAA only represent an estimate of the non-radiative recombination losses.”

2. Why did you use a repetition condition of 1 MHz of incident pulsed light for the TRPL measurement? If the light is irradiated before carrier recombination is complete, errors will occur in the carrier lifetime analytical results ($\tau \gg 1 \mu\text{s}$).

Response: We thank the reviewer for spotting this error in the manuscript. This was a mistake in reporting the experimental conditions. Indeed, for such long lifetimes we could not use a repetition rate of 1 MHz, but we actually had to use a repetition rate of 0.125 KHz. We have changed this in the main text.

3. error: Fig.5g (Description of the horizontal axis of the second graph) Thickness [eV] Thickness [μm]

Response: Thank you to the reviewer for spotting this error. We have corrected this in the updated version.

4. error : at lines 593, 523 nm CW laser  532 nm CW laser.

Response: Thank you to the reviewer for spotting this error. We have corrected this in the manuscript.

REVIEWERS' COMMENTS

Reviewer #1 (Remarks to the Author):

The authors have addressed the questions accordingly and the work can be published as is.

Reviewer #2 (Remarks to the Author):

The authors have revised the manuscript to take into account the reviewers' comments, especially the effect of tail states on non-radioactive recombination. The discussion is clear and the authors may publish it as is.